# (GIGA)bYte

DATA RELEASE

# The genome of *Tripterygium wilfordii* and characterization of the celastrol biosynthesis pathway

Tianlin Pei[1,2,†], Mengxiao Yan[1,†], Yu Kong[1], Hang Fan[1,2], Jie Liu[1], Mengying Cui[1], Yumin Fang[1], Binjie Ge[1], Jun Yang[1,2,*] and Qing Zhao[1,2,*]

1 Shanghai Key Laboratory of Plant Functional Genomics and Resources, Shanghai Chenshan Botanical Garden, Shanghai Chenshan Plant Science Research Center, Chinese Academy of Sciences, Shanghai, China

2 State Key Laboratory of Plant Molecular Genetics, CAS Center for Excellence in Molecular Plant Sciences, Shanghai Institute of Plant Physiology and Ecology, Chinese Academy of Sciences, Shanghai, China

## ABSTRACT

*Tripterygium wilfordii* is a vine from the Celastraceae family that is used in traditional Chinese medicine (TCM). The active ingredient, celastrol, is a friedelane-type pentacyclic triterpenoid with putative roles as an antitumor, immunosuppressive, and anti-obesity agent. Here, we report a reference genome assembly of *T. wilfordii* with high-quality annotation using a hybrid sequencing strategy. The total genome size obtained is 340.12 Mb, with a contig N50 value of 3.09 Mb. We successfully anchored 91.02% of sequences into 23 pseudochromosomes using high-throughput chromosome conformation capture (Hi–C) technology. The super-scaffold N50 value was 13.03 Mb. We also annotated 31,593 structural genes, with a repeat percentage of 44.31%. These data demonstrate that *T. wilfordii* diverged from Malpighiales species approximately 102.4 million years ago. By integrating genome, transcriptome and metabolite analyses, as well as *in vivo* and *in vitro* enzyme assays of two cytochrome P450 (CYP450) genes, *TwCYP712K1* and *TwCYP712K2*, it is possible to investigate the second biosynthesis step of celastrol and demonstrate that this was derived from a common ancestor. These data provide insights and resources for further investigation of pathways related to celastrol, and valuable information to aid the conservation of resources, as well as understand the evolution of Celastrales.

**Subjects** Genetics and Genomics, Botany, Plant Genetics

**Submitted:** 10 October 2020

\* Corresponding authors. E-mail: yangjun@csnbgsh.cn; zhaoqing@cemps.ac.cn

† Contributed equally.

Preprint submitted at https://doi.org/10.1101/2020.06.29.176958

## INTRODUCTION

*Tripterygium wilfordii* Hook. f. (NCBI: txid458696) is a perennial twining shrub belonging to the Celastraceae family. It is known in China as 'Lei gong teng' (meaning: Thunder God Vine). It is indigenous to Southeast China, the Korean Peninsula, and Japan, and has been cultivated worldwide as a medicinal plant [1, 2] (Figure 1). The extract of *T. wilfordii* bark has been used as a pesticide in China since ancient times, and was first recorded in the Illustrated Catalogues of Plants published in 1848 [3]. The potential medicinal activity of *T. wilfordii* has been studied since the 1960s, with its root being used to alleviate the symptoms of leprosy patients in Gutian County, Fujian Province, China [4]. This application ignited the interest of researchers in various fields. *T. wilfordii* was then reported to be effective in the treatment of autoimmune diseases, such as rheumatoid arthritis and systemic psoriasis [5, 6]. In recent decades, many studies have examined the potential anticancer, antidiabetic and anti-inflammatory effects of extracts of *T. wilfordii* [7–9].

Investigations into the pharmacological activities of *T. wilfordii* have mainly focused on the various compounds accumulating in its root, such as alkaloids, diterpenoids and triterpenoids [10, 11]. Celastrol is a friedelane-type triterpenoid that is mainly found in the root bark of *T. wilfordii* [12]. In Chinese medicine, it has been used for the treatment of inflammatory and autoimmune diseases [13], tumors [14], and as a possible treatment for Alzheimer's disease [15]. Celastrol is also a leptin sensitizer and may be useful in the treatment of obesity [16, 17]. Despite the commercial importance of natural products found in *T. wilfordii* and the growing demand for these products, traditional methods of production are becoming unsustainable owing to the slow growth rate of the vines and low accumulation of celastrol [18]. There is therefore a need for novel production methods, such as synthetic biological methods. Genome sequencing will provide a reference for mining the genes involved in the pathways of these bioactive compounds.

Celastrol is a pentacyclic triterpenoid synthesized from 2,3-oxidosqualene, the common biosynthetic precursor of triterpenoids derived from the cytosolic mevalonate (MVA) and plastid 2-C-methyl-D-erythritol-4-phosphate (MEP) pathways [19, 20]. Two oxidosqualene cyclases (OSCs), namely, TwOSC1 and TwOSC3, were identified as key enzymes in the cyclization of 2,3-oxidosqualene to form friedelin, the first step in celastrol formation [21]. The next step in this pathway is thought to be hydroxylation of the C-29 position of friedelin to produce 29-hydroxy-friedelin-3-one. This is then converted, via carboxylation, to polpunonic acid, which in turn undergoes a series of oxidation reactions and rearrangements to produce celastrol [21].

Here, we report the reference genome assembly of *T. wilfordii* using a combined sequencing strategy. After integrating genome, transcriptome and metabolite analyses, several novel cytochrome P450 (CYPs) proteins related to celastrol biosynthesis were identified. TwCYP712K1 and TwCYP712K2 were then functionally characterized using *in vivo* yeast and *in vitro* enzyme assays. These data represent a strategy to reveal the evolution of Celastrales and the key genes involved in celastrol biosynthesis.

## METHODS

A protocol collection including methods for DNA-extraction, Hi–C assembly and optical mapping is available via protocols.io (Figure 2) [22].

### Plant materials

*Tripterygium wilfordii* plants were collected from the experimental fields of Shanghai Chenshan Botanical Garden (31° 04′ 30.00′′ N, 121° 10′ 58.93′′ E) and cultured in a greenhouse by cutting propagation. All materials used for genome sequencing originated from a single plant (grown in the greenhouse of our laboratory, voucher TW1). For RNA sequencing (RNA-seq), tissues from roots (R), stems (S), young leaves (YL), mature leaves (L), flower buds (FB) and flowers (F) were harvested with three independent biological replicates.

### DNA sequencing

Total DNA was isolated from leaves using the modified cetyltrimethylammonium bromide (CTAB) method [23]. DNA purity was checked by electrophoretic analysis on a 1% agarose gel and using a NanoPhotometer spectrophotometer (IMPLEN, CA, USA). The DNA concentration was determined using a Qubit 2.0 fluorometer (Life Technologies, CA, USA).

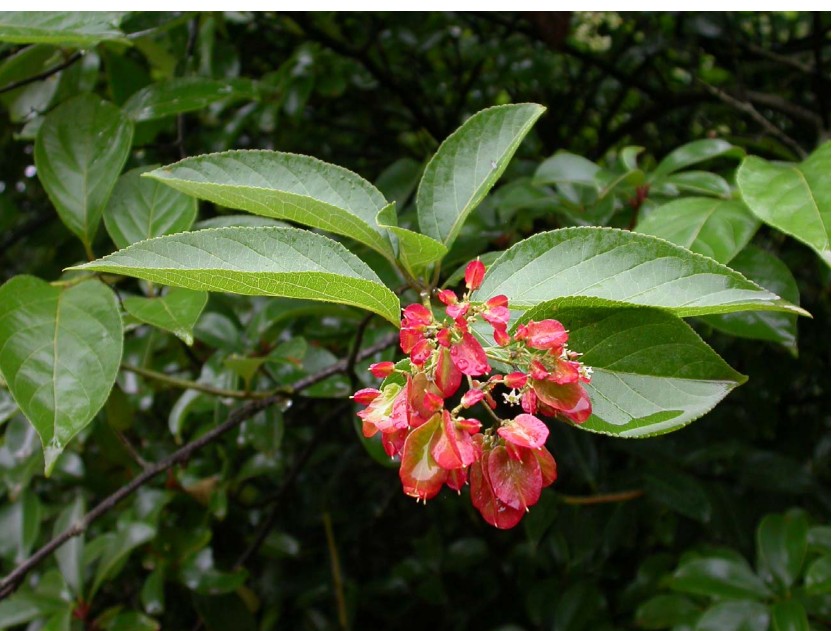

**Figure 1.** Picture of *Tripterygium wilfordii*. With thanks to Dr. Bin Chen from the Shanghai Chenshan Herbarium for providing the image.

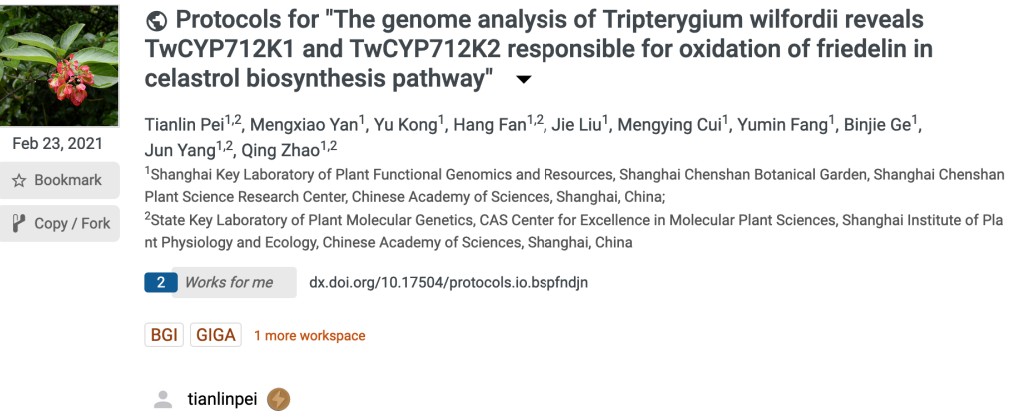

**Figure 2.** Protocol collection for the genome analysis of *Tripterygium wilfordii*. https://www.protocols.io/widgets/doi?uri=dx.doi.org/10.17504/protocols.io.bspfndjn

For Illumina sequencing, qualified DNA was fragmented using a Covaris device (MA, USA). Fragmented DNA was end-repaired; poly(A) tail and adaptor addition was performed using the Next Ultra DNA Library Prep Kit (NEB, MA, USA), then the appropriate samples were selected by electrophoretic analysis. The size-selected product was PCR-amplified, and the final product was purified and validated using AMPure XP beads (Beckman Coulter, CA, USA) and an Agilent Bioanalyzer 2100. Using the HiSeq 2500 platform, 150-bp paired-were sequenced. Clean data were obtained by removing adaptor reads, unidentified nucleotides (N) and low-quality reads from the raw reads, and Q20, Q30 and GC content of the clean data were calculated for quality assessment (Table 1). We estimated the genome size by

| Raw paired reads | Raw Base (bp) | Effective Rate (%) | Error Rate (%) | Q20 (%) | Q30 (%) | GC Content (%) |
|---|---|---|---|---|---|---|
| 84,395,810 | 25,318,743,000 | 99.69 | 0.05 | 95.44 | 88.88 | 38.22 |

**Table 1.** Genome sequencing data and sequencing coverage.

performing k-mer frequency analysis. The k-mer frequencies (k-mer size = 17) were obtained using Jellyfish v2.2.7 [24] with jellyfish count -G 2 -s 5G -m 17 and jellyfish stats as the default parameters.

For long-read sequencing, qualified DNA was sheared into fragments in a g-TUBE (Covaris, MA, USA) by centrifugation, and quantity and quality were controlled by an Agilent Bioanalyzer 2100. To construct a sequencing library, the fragmented DNA was end-repaired and poly(A) tail and adaptor addition was performed using Next Ultra II End Repair/dA-Tailing Module, Next FFPE DNA Repair Mix and Next Quick Ligation Module (NEB, MA, USA), respectively, according to the manufacturer's instructions. The final product was validated using an Agilent Bioanalyzer 2100. Finally, the qualified DNA library was sequenced using Oxford Nanopore Technology (ONT) on the PromethION platform.

## Genome assembly

*De novo* genome assembly was carried out using NextDenovo v2.3.0 [25]. The correct_option parameters used were: read_cutoff = 1k, seed_cutoff = 28087, pa_correction = 20, seed_cutfiles = 100, sort_options = -m 15g -t 8 -k 40, minimap2_options_raw = -x ava-ont -t 8. The assemble_option parameters used were: random_round = 20, minimap2_options_cns = -x ava-ont -t 8 -k17 -w17, nextgraph_options = -a 1.

Racon v1.3.1 [26] and Pilon v1.22 (Pilon, RRID:SCR_014731) [27] were used for error correction with ONT data and Illumina data, respectively. Error correction was performed three times with default parameters. The completeness of the genome assembly was assessed using Benchmarking Universal Single-Copy Orthologs (BUSCO) v3.0.2 (BUSCO, RRID:SCR_015008) [28] with the parameters: -m genome -c 15 -sp arabidopsis. Assembly accuracy was evaluated using Burrows–Wheeler Aligner (BWA) software (version: 0.7.8-r455) [29] to align Illumina reads back to the genome. Variant calling was performed using SAMtools (version: 0.1.19-44428cd, SAMTOOLS, RRID:SCR_002105) [30, 31] with parameters: -m 2 -F 0.002 -d 110 -u -f. To assess the genome assembly quality, transcriptome data were assembled using Trinity (Trinity, RRID:SCR_013048) [32], then mapped back to the scaffolds using BLAT (BLAT, RRID:SCR_011919) [33].

For Bionano sequencing, genomic DNA (molecules > 300 kb) of leaves from living *T. wilfordii* plants was extracted using the Plant DNA Isolation Kit (Bionano Genomics, CA, USA). Using the NLRS DNA Labeling Kit (Bionano Genomics, CA, USA), DNA molecules were digested with Nt.BspQI endonucleases (determined after evaluation by electronic digestion), and fluorescently labeled. Labeled DNA molecules were electrophoretically stretched into linearization by Saphyr Chip (Bionano Genomics, CA, USA), passed through the NanoChannels [34], and then captured on the Saphyr platform with a high-resolution camera. Raw image data were first converted to digital representations of the motif-specific label pattern, then analyzed using Bionano Solve v3.1 [35] and its in-house scripts. Bionano data were compared with the draft genome (Nanopore version) with the parameters: -U, -d, -T, 3, -j, 3, -N, 20, -I, 3, and scaffolds were generated by connecting contigs with the parameters: -f, -B, 1, -N, 1.

**Table 2.** Statistics of genome assembly.

| Raw bases (bp) | Clean bases (bp) | Effective Rate(%) | Error rate | Q20 (%) | Q30 (%) | GC content (%) |
|---|---|---|---|---|---|---|
| 78,004,410,900 | 77,678,361,000 | 99.58 | 0.04 | 96.66 | 91.13 | 39.83 |

## Sequence anchoring

Hi–C library preparation and sequencing were based on a protocol described previously, with some modifications [36–38]. Leaves from living *T. wilfordii* plants were treated with 1% formaldehyde solution to fix chromatin. Approximately 2 g of fixed tissue was homogenized with liquid nitrogen, resuspended in nucleus isolation buffer and filtered with a 40-nm cell strainer. Extracted chromatin was cut with the HindIII restriction enzyme (NEB, MA, USA), end-filled, then labeled with biotin. After ligation with T4 DNA ligase (NEB, CA, USA) and reversal of crosslinking by proteinase K, DNA was purified, cleaved into 350-bp fragments and end-repaired. DNA fragments labeled with biotin were separated using Dynabeads M-280 Streptavidin (Thermo Fisher, MA, USA), purified, and end-repaired. A-tails were added and adaptors were ligated, and the sequences were amplified by PCR to generate Hi–C libraries. Finally, the qualified libraries were sequenced on an Illumina platform. Clean data were obtained by removing adaptor reads, unidentified nucleotides (N) and low-quality reads from the raw reads, and Q20, Q30 and GC content of the clean data were calculated for quality assessment (Table 2). Clean data were first mapped to the draft genome using BWA software (version: 0.7.8-r455) [29]. After removal of PCR duplicates and unmapped reads using SAMtools v1.9 [39], based on the numbers of interacting read pairs, contigs were clustered and ordered into chromosome groups using LACHESIS (version 201701) [40] with the parameters: RE_SITE_SEQ = GATC, CLUSTER_N = 23, CLUSTER_CONTIGS_WITH_CENS = −1, CLUSTER_MIN_RE_SITES = 388, CLUSTER_MAX_LINK_DENSITY = 3, CLUSTER_NONINFORMATIVE_RATIO = 0, CLUSTER_DRAW_HEATMAP = 1, and CLUSTER_DRAW_DOTPLOT = 1.

## Transcriptome sequencing

Total RNA was extracted from collected tissues using the RNAprep Pure Plant Kit (TIANGEN, Beijing, China). Qualified RNA from each sample was used to generate sequencing libraries using the NEBNext Ultra RNA Library Prep Kit for Illumina sequencing (NEB, MA, USA), following the manufacturer's instructions. mRNA was purified from total RNA using poly(T) oligo-attached magnetic beads, then cleaved into short fragments that were used as templates for cDNA synthesis. After purification, repair, adenylation and adaptor ligation of the 3′ end, 150 to 200-bp cDNA fragments were separated for PCR amplification. Finally, libraries were quality controlled using an Agilent 2100 Bioanalyzer and qPCR, then sequenced on the Illumina HiSeq 2500 platform. Raw reads with adapter, poly(N) and low-quality reads were removed to generate clean data, and Q20, Q30 and GC content of the clean data were calculated for quality assessment (Table 3). RNA-seq data was mapped back to the genome assembly of *T. wilfordii* using HISAT v2.0.4 with default parameters [41]. The read numbers of each gene were counted using HTSeq v0.6.1 with the parameter: –m union [42]. Fragments per kilobase of transcript per million fragments mapped (FPKM) of each gene was calculated based on the length of the gene and number of read counts mapped to this gene [43].

**Table 3.** Nucleotide statistics in the draft genome assembly.

| Sample name | Raw reads (bp) | Clean reads (bp) | Clean bases | Error rate (%) | Q20 (%) | Q30 (%) | GC content (%) |
|---|---|---|---|---|---|---|---|
| R1 | 68,286,484 | 67,508,000 | 10.13G | 0.03 | 98.00 | 94.11 | 45.48 |
| R2 | 58,613,186 | 57,798,058 | 8.67G | 0.03 | 98.00 | 94.09 | 46.14 |
| R3 | 63,088,548 | 61,991,218 | 9.3G | 0.02 | 98.05 | 94.22 | 45.28 |
| YL1 | 61,340,056 | 60,217,856 | 9.03G | 0.03 | 98.02 | 94.11 | 45.98 |
| YL2 | 41,099,134 | 40,542,280 | 6.08G | 0.02 | 98.28 | 94.94 | 45.82 |
| YL3 | 69,124,902 | 68,072,478 | 10.21G | 0.02 | 98.11 | 94.35 | 45.60 |
| L1 | 66,626,224 | 65,459,234 | 9.82G | 0.02 | 98.12 | 94.34 | 45.39 |
| L2 | 58,487,674 | 57,306,282 | 8.6G | 0.02 | 98.30 | 94.76 | 45.35 |
| L3 | 70,363,762 | 69,393,172 | 10.41G | 0.02 | 98.14 | 94.41 | 45.88 |
| S1 | 55,910,376 | 55,176,316 | 8.28G | 0.02 | 98.10 | 94.27 | 44.96 |
| S2 | 67,911,592 | 66,929,018 | 10.04G | 0.03 | 98.02 | 94.11 | 44.93 |
| S3 | 59,810,518 | 58,957,160 | 8.84G | 0.03 | 97.97 | 93.94 | 44.95 |
| FB1 | 64,749,004 | 63,837,752 | 9.58G | 0.03 | 98.01 | 94.06 | 45.30 |
| FB2 | 53,079,594 | 52,523,514 | 7.88G | 0.03 | 98.01 | 94.14 | 45.77 |
| FB3 | 53,084,804 | 52,578,262 | 7.89G | 0.02 | 98.08 | 94.30 | 45.76 |
| F1 | 63,007,696 | 62,341,814 | 9.35G | 0.02 | 98.14 | 94.38 | 44.91 |
| F2 | 60,886,358 | 60,100,754 | 9.02G | 0.02 | 98.13 | 94.36 | 45.68 |
| F3 | 59,719,778 | 59,080,108 | 8.86G | 0.03 | 97.97 | 93.99 | 44.95 |

R, root; YL, young leaf; L, mature leaf; S, stem; FB, flower bud; F, flower. Numbers (1-3) represent three replicates, respectively.

For full-length transcriptome sequencing by PacBio, the best quality RNA samples of each tissue were mixed together to build an isoform sequencing library using the Clontech SMARTer PCR cDNA Synthesis Kit and the BluePippin Size Selection System protocol, as described by Pacific Biosciences (PN 100-092-800-03). Samples were then sequenced on the PacBio Sequel platform. Sequence data were processed using SMRTlink 7.0 software [44] with the parameters: –minLength 50, –maxLength 15000, –minPasses 1. Error correction was achieved using the Illumina RNA-seq data with LoRDEC v0.7, with the parameters: -k 23, -s 3 [45]. Redundancy in the corrected consensus reads was removed by CD-HIT v4.6.8 [46], with the parameters: -c 0.95, -T 6, -G 0, -aL 0.00, -aS 0.99, -AS 30 to obtain the final transcripts for the subsequent analysis.

## Genome annotation

Homolog alignment and *de novo* prediction were applied for repeat annotation. For homolog alignment, the Repbase database employing RepeatMasker software v4.0.7 (RepeatMasker, RRID:SCR_012954) and its in-house scripts (RepeatProteinMask v4.0.7) was used with default parameters to extract repeat sequences [47]. For *de novo* prediction, LTR_FINDER v1.0.7 (LTR_Finder, RRID:SCR_015247) [48], RepeatScout v1.0.5 (RepeatScout, RRID:SCR_014653) [49], and RepeatModeler v1.0.3 (RepeatModeler, RRID:SCR_015027) [50] were used with default parameters to build a *de novo* repetitive element database for repeat identification. Tandem repeats were also extracted by *de novo* prediction using TRF v4.0.9 [51].

A combined strategy based on homology, gene prediction, RNA-seq and PacBio data was used to annotate gene structure. For homolog prediction, sequences of proteins from six species, including *Arabidopsis thaliana*, *Vitis vinifera*, *Medicago truncatula*, *Cucumis sativus*, *Ricinus communis*, and *Glycyrrhiza uralensis*, were downloaded from Ensembl/National Center for Biotechnology Information (NCBI)/DNA Database of Japan (DDBJ). Protein

sequences were aligned to the genome using TblastN v2.2.26 [52] (*E*-value ≤ 1 × 10$^{-5}$), then the matching proteins were aligned to homologous genome sequences for accurate spliced alignments with GeneWise v2.4.1 (GeneWise, RRID:SCR_015054) [53]. *De novo* gene structure identification was based on Augustus v3.2.3 (Augustus, RRID:SCR_008417) [54], GlimmerHMM v3.04 (GlimmerHMM, RRID:SCR_002654) [55], and SNAP (2013-11-29) [56]. Based on the above prediction results, RNA-seq reads from different tissues, and PacBio reads, were aligned to the genome using HISAT v2.0.4 (HiSat2, RRID:SCR_015530) [41] and TopHat v2.0.12 (TopHat, RRID:SCR_013035) [57] with default parameters to identify exon regions and splice positions. Alignment results were then used as input for Stringtie v1.3.3 (StringTie, RRID:SCR_016323) [58] with default parameters for genome-based transcript assembly. Alignment results were then integrated into a nonredundant gene set using EVidenceModeler v1.1.1 and further corrected with Program to Assemble Spliced Alignment (PASA) to predict untranslated regions and alternative splicing to generate the final gene set [59].

According to the final gene set, gene function was predicted by aligning the protein sequences to Swiss-Prot [60] and the Non-Redundant Protein Sequence Database (NR) (version 20190709) [61]. The motifs and domains were annotated using InterProScan70 v5.31 by searching against the Protein Families Database (Pfam) [62], Kyoto Encyclopedia of Genes and Genomes (KEGG, version 20190601) [63], and Integrative Protein Signature Database (InterPro) v32.0 [64] using Blastp (*E*-value ≤ 1 × 10$^{-5}$). Gene Ontology (GO) IDs for each gene were assigned according to the corresponding InterPro entry.

Noncoding RNA was annotated using tRNAscan-SE v1.4 (for tRNA) [65] or INFERNAL v1.1.2 with default parameters (for miRNA and snRNA) [66]. rRNA was predicted by BLAST using rRNA sequences from *A. thaliana* and *O. sativa* as references, which are highly conserved among plants.

## Comparative genome analyses

Gene family clustering of 12 species, including *T. wilfordii*, *A. thaliana*, *Citrus sinensis*, *V. vinifera*, *Glycine max*, *M. truncatula*, *G. uralensis*, *C. sativus*, *Populus trichocarpa*, *R. communis*, *Oryza sativa*, and *Amborella trichopoda*, was inferred through all-against-all protein sequence similarity searches using OthoMCL v1.4 [67], with the parameters: -mode 3 and -inflation 1.5. Proteins containing fewer than 50 amino acids were removed, and only the longest predicted transcript per locus was retained.

Single-copy orthologous genes were retrieved from the 12 species and aligned using MUSCLE v3.8.31 with default parameters [68]. All alignments were combined to produce a super-alignment matrix, which was used to construct a maximum likelihood (ML) phylogenetic tree using RAxML v8.2.12 [69] with the parameters: cds: -m GTRGAMMA -p 12345 -x 12345 -#100 -f ad -T 20, pep: -m PROTGAMMAAUTO -p 12345 -x 12345 -#100 -f ad -T 20.

Divergence times between species were calculated using the MCMCtree v4.9 program implemented for phylogenetic analysis by maximum likelihood (PAML) with the default parameters [70]. The following calibration points were applied: *M. truncatula–G. uralensis* (15–91 million years ago, Mya), *G. max–M. truncatula* (46–109 Mya), *G. max–C. sativus* (95–135 Mya), *A. thaliana–C. sinensis* (96–104 Mya), *P. trichocarpa–R. communis* (70–86 Mya), *A. thaliana–P. trichocarpa* (98–117 Mya), *C. sativus–R. communis* (101–131 Mya), *V. vinifera–A.*



**Table 4.** Primers used for gene cloning.

| Primer names | Sequence (5′ to 3′)* |
|---|---|
| TwCYP712K1-F | GGGGACAAGTTTGTACAAAAAAGCAGGCTTCATGGCCACCATCACTGACATC |
| TwCYP712K1-R | GGGGACCACTTTGTACAAGAAAGCTGGGTTTTAACCGGCAAATGGATTGAA |
| TwCYP712K2-F | GGGGACAAGTTTGTACAAAAAAGCAGGCTTCATGACAACAATCACTGATGTGAA |
| TwCYP712K2-R | GGGGACCACTTTGTACAAGAAAGCTGGGTTTTAAGAAGAAAATGGATTGAACC |
| TwCYP712K3-F | GGGGACAAGTTTGTACAAAAAAGCAGGCTTCATGGCCACCACTACCATCATT |
| TwCYP712K3-F | GGGGACCACTTTGTACAAGAAAGCTGGGTTTTAGCAAGAAAAGGGATGGAATC |

*\* Underlined sequences represent Gateway prime.*

*thaliana* (107–135 Mya), *V. vinifera–O. sativa* (115–308 Mya), and *O. sativa–A. trichopoda* (173–199 Mya). These calibrations were extracted from TimeTree [71].

Expansion and contraction of gene families were analyzed by using CAFÉ v4.2 [72] with the parameters: -p 0.05 -t 4 -r 10000. To avoid false positives, results were filtered and the enrichment results screened with a family-wide *P*-value < 0.05 and Viterbi *P*-values < 0.05.

### Genome-wide identification of CYP genes

The hidden Markov model (HMM) profile of Pfam PF06200 [62] was used to extract full-length CYP candidates from the *T. wilfordii* genome by the HMM algorithm (HMMER) [71], filtering by a length between 400 and 600 amino acids [74].

### Phylogenetic analyses

Multiple sequence alignments and phylogenetic tree construction were performed using MEGA X [75], with either the neighbor-joining or ML method with a bootstrap test (*n* = 1000 replications).

### Co-expression analysis

Gene expression pattern analysis was performed using Short Time-series Expression Miner (STEM) software [76] on the OmicShare tools platform [77]. The parameters were set as follows: the maximum unit change in model profiles between time points was 1; the maximum output profile number was 20 (similar profiles were merged); the minimum ratio of fold change of differentially expressed genes (DEGs) was no less than 2.0, and the *P*-value was <0.05.

### Gene cloning

The complete open reading frames (ORFs) of the putative CYP genes were amplified using the primers listed in Table 4, with cDNA from *T. wilfordii* root used as the template. According to the manufacturer's instructions, fragments were cloned into the entry vector pDONR207 and yeast expression vector pYesdest52 using the Gateway BP Clonase II Enzyme Kit and LR Clonase II Enzyme Kit (Invitrogen, MA, USA), respectively.

### Standard compounds

Friedelin, 29-hydroxy-friedelan-3-one, and celastrol were purchased from Yuanye-Biotech (Shanghai, China), and polpunonic acid and wilforic acid A were purchased from Weikeqi-Biotech (Sichuan, China). Friedelin was dissolved in dimethyl sulfoxide (DMSO)/isopropanol (v/v = 1:2) following 30 min of ultrasonication in a water bath, while

29-hydroxy-friedelan-3-one, celastrol, polpunonic acid and wilforic acid A were dissolved in methanol.

## Metabolite analysis

Plant tissue was ground into powder in liquid nitrogen then freeze dried. Fifty milligrams of sample was suspended in 2 mL of 80% (v/v) methanol, set overnight at room temperature, then extracted in an ultrasonic water bath for 60 min. After centrifugation at 12,000$g$ for 2 min, the supernatant was filtered through a 0.2-$\mu$m Millipore filter before liquid chromatography–mass spectrometry (LC–MS) analysis.

Levels of celastrol and wilforic acid A were analyzed using an Agilent 1260LC-6400 QQQ (triple quadrupole mass spectrometer). Chromatographic separation was carried out on an Agilent Eclipse XDB-C18 analytical column (4.6 × 250 mm, 5 $\mu$m) with a guard column. The flow rate of the mobile phase consisting of 0.1% (v/v) formic acid in water (A) and acetonitrile (B) was set to 0.8 mL/min. The gradient program was as follows: 0–12 min, 10–60% B; 12–17 min, 70% B; 17–25 min, 95% B; 25–28 min, 95% B; 28–29 min, 5% B; 29–35 min, 5% B. The detection wavelength of celastrol was 425 nm, and UV spectra from 190–500 nm were also recorded. The injection volume was 10 $\mu$l and the column temperature was 35 °C. The liquid chromatography (LC) effluent was introduced into the electrospray ionization (ESI) source by a split-flow valve with a ratio of 3:1. All mass spectra were acquired in negative ion mode, and the parameters were as follows: drying gas 4 L/min; drying gas temperature 300 °C; nebulizer (high-purity nitrogen) pressure 15 psi; capillary voltage 4.0 kV; fragmentor voltage 135 V; and cell accelerator voltage 7 V. For full-scan mass spectrometry (MS) analysis, the spectra were recorded in the m/z range of 100–750.

Levels of 29-hydroxy-friedelan-3-one and polpunonic acid were analyzed using Thermo Q Exactive Plus. Chromatographic separation was carried out on a Thermo Syncronis C18 column (2.1 × 100 mm, 1.7 $\mu$m). The flow rate of the mobile phase, consisting of 0.1% (v/v) formic acid in water (A) and acetonitrile (B), was set to 0.4 mL/min. The gradient program was as follows: 0–12 min, 10–60% B; 12–17 min, 70% B; 17–25 min, 95% B; 25–28 min, 95% B; 28–29 min, 5% B. Mass spectra were acquired in both positive and negative ion modes with a heated ESI source, and the parameters were as follows: aus. gas flow 10 L/min; aus. gas heater 350 °C; sheath gas flow 40 L/min; spray voltage 3.5 kV; capillary temperature 320 °C. For full-scan MS/data-dependent (ddMS$^2$) analysis, spectra were recorded in the m/z range of 50–750 at a resolution of 17,500 with automatic gain control (AGC) targets of 1 × 10$^6$ and 2 × 10$^5$, respectively. Levels of metabolites in different tissues were measured by comparing the area of the individual peaks with standard curves obtained from standard compounds.

## Enzyme assays of yeast *in vivo*

Yeast *in vivo* assays were performed following a previously described protocol with some modifications [78]. The yeast expression vector constructs or empty vector were transformed into the yeast *Saccharomyces cerevisiae* WAT11 [79, 80] using the Yeast Transformation II Kit (ZYMO, CA, USA), and screened on synthetic-dropout (SD) medium lacking uracil (SD-Ura) with 20 g/L glucose. After growing at 28 °C for 48–72 h, transformant colonies were initially grown in 20 ml of SD-Ura liquid medium with 20 g/L glucose at 28 °C for approximately 24 h until the OD$_{600}$ reached 2–3. Yeast cells were harvested by centrifugation at 4000$g$ and resuspended in 20 mL of SD-Ura liquid medium supplemented with 20 g/L galactose to induce target proteins, while friedelin or



| Table 5. Genome sequencing data and sequencing coverage. | | |
|---|---|---|
| **Pair-end libraries** | **Total data (Gb)** | **Sequence coverage (X)** |
| Illumina Hiseq PE150 (for genome survey and error correction) | 25.32 | 67.37 |
| Nanopore | 77.86 | 207.16 |
| BioNano | 60.80 | 161.77 |
| Illumina Hiseq PE150 (for Hi–C) | 77.68 | 206.68 |
| PacBio (for annotation) | 20.75 | 55.21 |

29-hydroxy-friedelane-3-one was applied to the cultures at a final concentration of 25 mM. After 48 h of fermentation (supplemented with 2 mL galactose after 24 h), yeast cells were harvested by centrifugation and extracted with 2 mL of 70% methanol in an ultrasonic water bath for 2 h. The supernatants were filtered with a 0.2-μm Millipore filter and analyzed by LC–MS.

### Enzyme assays *in vitro*

The protocol for enzyme assays *in vitro* was performed as described previously with some modifications [81]. Yeast transformation and target protein induction were performed as described above, except for 24 h of fermentation after galactose supplementation. Yeast cells were harvested by centrifugation and suspended in a 10-mL mixture of 50 mM Tris-HCl (pH 7.5), 1 mM EDTA, 0.5 mM phenylmethylsulfonyl fluoride, 1 mM dithiothreitol, 0.6 M sorbitol and ddH$_2$O. High pressure cell disruption equipment (Constant Systems, Northants, UK) was used to crush the yeast cells. After centrifugation, approximately 10 mL of supernatant was collected, and CaCl$_2$ was applied at a final concentration of 18 mM. Microsomal proteins were then collected by centrifugation and suspended in storage buffer containing 50 mM Tris-HCl (pH 7.5), 1 mM EDTA and 20% (v/v) glycerol with a final concentration of 10–15 mg/mL determined by the Bradford method [82].

The catalytic activity of putative CYP was assayed in a 100-μl reaction volume, which contained 100 mM sodium phosphate buffer (pH 7.9), 0.5 mM reduced glutathione, 2.5 μg of extracted protein and 100 μM substrate (friedelin or 29-hydroxy-friedelan-3-one). The reaction was initiated by adding NADPH at 1 mM and incubating for 12 h at 28 °C. Methanol was then added at a final concentration of 70% to quench the reaction. The reaction mixture was filtered with a 0.2-μm Millipore filter and analyzed by LC–MS. Microsomal proteins extracted from yeast harboring the empty vector were used as a negative control.

### Syntenic analyses

The genomes of *T. wilfordii*, *O. sativa* japonica and *V. vinifera* were compared using MCScan Toolkit v1.1 [83] implemented in Python. The genomes of *O. sativa* v32 and *V. vinifera* v32 were downloaded from Ensembl Plants [84]. Syntenic gene pairs were identified using an all-vs-all BLAST search using LAST [85], filtered to remove pairs with scores below 0.7, and clustered into syntenic blocks in MCScan. Microsynteny plots were constructed using MCScan.

## RESULTS

## Genome sequencing, assembly, and annotation

We obtained 77.86 Gb of Nanopore reads, amounting to 207.16× coverage of the 375.84-Mb genome, a size estimated by k-mer distribution analysis (Figure 3 and Table 5).



**Table 6.** Statistics of genome assembly.

| Sample ID | Nanopore version | | | | Bionano hybrid scaffold version | | | |
|---|---|---|---|---|---|---|---|---|
| | Length | | Number | | Length | | Number | |
| | Contig* (bp) | Scaffold (bp) | Contig* | Scaffold | Contig* (bp) | Scaffold (bp) | Contig* | Scaffold |
| Total | 340,124,188 | - | 553 | - | 340,124,188 | 342,588,429 | 553 | 470 |
| Max | 7,962,777 | - | - | - | 7,962,777 | 10,510,391 | - | - |
| Number ≥ 2000 | - | - | 553 | - | - | - | 553 | 470 |
| N50 | 3,088,446 | - | 34 | - | 3,088,446 | 5,425,714 | 34 | 25 |
| N60 | 2,351,287 | - | 46 | - | 2,351,287 | 4,027,572 | 46 | 32 |
| N70 | 1,048,940 | - | 68 | - | 1,048,940 | 3,189,075 | 68 | 41 |
| N80 | 334,278 | - | 133 | - | 334,278 | 634,293 | 133 | 63 |
| N90 | 202,837 | - | 264 | - | 202,837 | 205,847 | 264 | 179 |

*Contigs after scaffolding.

**Table 7.** Nucleotide statistics in the draft genome assembly.

| | Number (bp) | % of genome |
|---|---|---|
| A | 106,752,681 | 31.39 |
| T | 106,872,067 | 31.42 |
| C | 63,093,649 | 18.55 |
| G | 63,405,791 | 18.64 |
| N | 0 | 0.00 |
| Total | 340,124,188 | - |
| GC | 126,499,440 | 37.19 |

**Table 8.** SNP statistics of the draft genome assembly.

| | Number | Percentage |
|---|---|---|
| All SNP | 766,560 | 0.256773% |
| Heterozygosis SNP | 756,672 | 0.253461% |
| Homology SNP | 9888 | 0.003312% |

The draft genome was assembled to obtain primary contigs, with a total size of 340.12 Mb and contig N50 of 3.09 Mb (Table 6).

The GC content of the genome was 37.19%, with 0.00% N (Table 7).

Variant calling showed a heterozygosity rate of 0.25% (Table 8).

BUSCO analysis showed 95.2% complete single-copy genes (Table 9).

Short reads obtained from Illumina sequencing in the genome survey were aligned to the genome (Table 5), which exhibited a high consistency with a 95.31% mapping rate and 93.99% coverage (Table 10).

In addition, 87.85% of expressed sequence tags (ESTs) could be identified in the assembly, indicating high coverage of the genome (Table 11).

For assembly improvement, 60.80 Gb (161.77× of estimated genome size) of reads from Bionano sequencing were obtained and integrated with the draft genome to construct scaffolds, which updated the genome of *T. wilfordii* from a total contig length of 340.12 Mb and a contig N50 of 3.09 Mb to a total scaffold length of 342.59 Mb and a scaffold N50 of 5.43 Mb (Tables 12 and 5).

Furthermore, we anchored 91.02% of the original 342.61-Mb assembly into 23 groups using Hi–C technology (Tables 13 and 14).

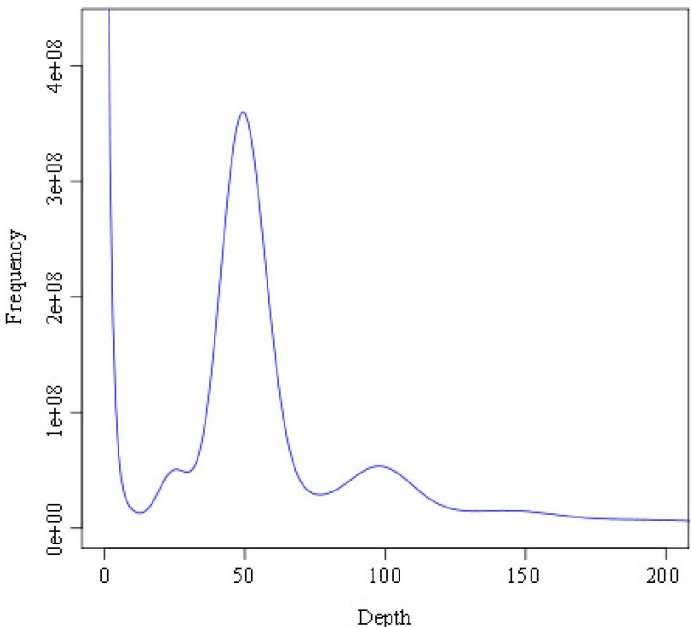

**Figure 3.** Estimation of *Tripterygium wilfordii* genome size by k-mer analysis. *X* axis shows k-mer depth and *Y* axis shows K-mer frequency. $G_0$ = k-mer number/depth, $G = G_0^*$ (1 − Error rate) where $G_0$ is previous genome size and *G* is revised genome size. The genome size was measured as 375.84 Mb using this method.

All the super-scaffold was able to be placed in one of 23 groups (Figure 4). The super-scaffold N50 reached 13.03 Mb, with the longest super-scaffold being 17.75 Mb in size (Tables 13 and 14). The number of groups, hereafter referred to as pseudochromosomes, corresponded well to the number of chromosomes reported previously [86].

For genome annotation and gene expression profile analyses, roots, stems, young leaves, mature leaves, flower buds and flowers of *T. wilfordii* plants were collected prior to RNA-seq using the Illumina platform. Furthermore, RNA samples from different tissues were mixed, then sequenced using the PacBio platform to obtain full-length transcriptome sequences (Table 5). A combined strategy involving *de novo* prediction, homology prediction, RNA-seq and PacBio read alignment was used to construct the gene structure for the *T. wilfordii* genome. The final set of annotated genes amounted to 31,593 genes, with an average length of 3180 bp and an average coding sequence length of 1182 bp (Table 15).

A total of 27,301 genes (86.41%) were supported by RNA-seq data and 23,229 genes (73.53%) were supported by all the methods used; these genes were annotated with high confidence. Gene function annotation was performed by BLAST analysis of the protein sequences of predicted genes against public databases, including NR, Swiss-Prot, KEGG, GO, Pfam and InterPro. A total of 30,535 (96.70%) gene products could be functionally predicted, and 22,491 sequences could be annotated by at least one of the databases (NR, SwissProt, InterPro and KEGG) (Table 16).

Repeat sequence annotation showed that the *T. wilfordii* genome contained 44.31% repetitive sequences. Among these sequences, tandem repeats (small satellites and microsatellites) and interspersed repeats accounted for 0.95% and 43.36%, respectively. Long terminal repeats (LTRs) of retroelements were the most abundant interspersed

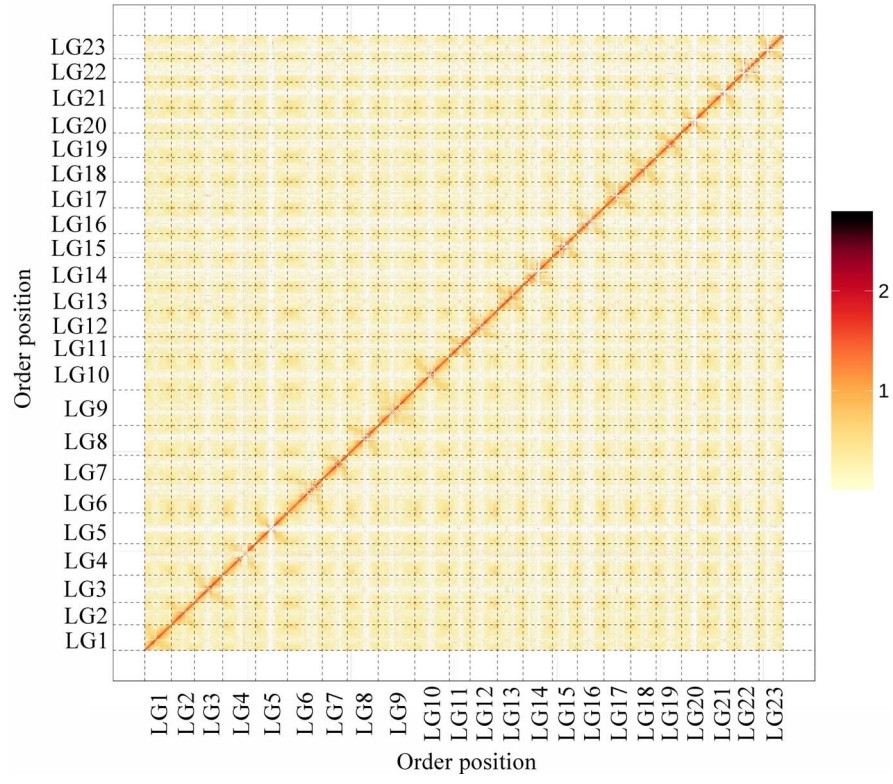

**Figure 4.** Interaction heat-map of chromosomal fragments based on Hi–C analysis. LG1–LG23 indicate Lachesis Groups 1-23. *X* and *Y* axes indicate the order positions of scaffolds on corresponding pseudochromosomes. The bar represents interaction strength between sequence segments.

**Table 9.** Summary of BUSCO evaluation.

|  | Percentage (%) |
|---|---|
| Complete BUSCOs | 95.2 |
| Complete and single-copy BUSCOs | 77.7 |
| Complete and duplicated BUSCOs | 17.5 |
| Fragmented BUSCOs | 1.1 |
| Missing BUSCOs | 3.7 |
| Total BUSCO groups searched | 1440 |

repeats, occupying 36.74% of the genome, including 13.70% *Gypsy* LTRs and 9.84% *Copia* LTRs, followed by DNA transposable elements at 1.68% (Table 17).

Noncoding RNA annotation revealed that the *T. wilfordii* genome possessed 355 microRNAs (miRNAs), 797 transfer RNAs (tRNAs), 827 ribosomal RNAs (rRNAs), and 982 small nuclear RNAs (snRNAs) (Table 18).

Integrated distributions of the genes, repeats, noncoding RNA densities, and all detected segmental duplications are shown in Figure 5.

## Comparative genomic analysis

To identify evolutionary characteristics and gene families, the *T. wilfordii* genome was compared with 11 published genomes of nine eudicot species (*A. thaliana*, *C. sinensis*,



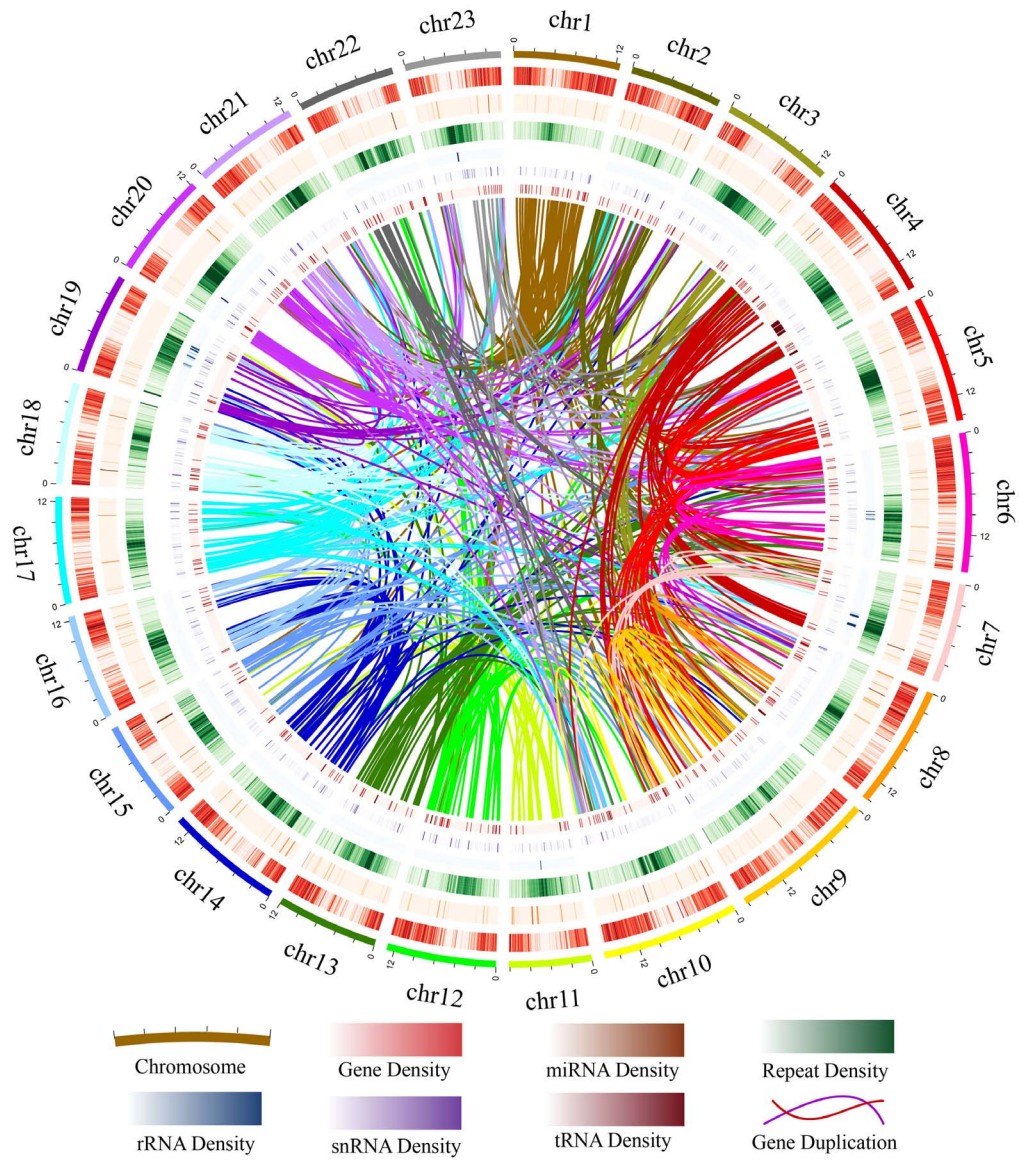

**Figure 5.** Landscape of the *Tripterygium wilfordii* genome assembly. The circles (outer to inner) represent: pseudochromosomes, gene density, miRNAs, repeats, rRNAs, snRNAs, tRNAs, and duplicated gene links within the genome. The scale shows chromosomes in a 500-kb window; gene density in a 100-kb window (0–100%, which means that the percentage of gene density indicated by the color gradient starts from 0 and goes to 100% of 100 kb of DNA); miRNA density in a 100-kb window (0–1%); repeat density in a 100-kb window (0–100%); rRNA density in a 100-kb window (0–18%); snRNA density in a 100-kb window (0–1.3%); tRNA density in a 100-kb window (0–1.8%); and detected gene duplication links (570).

*V. vinifera, G. max, M. truncatula, G. uralensis, C. sativus, P. trichocarpa* and *R. communis*) and a monocot species (*O. sativa*). In addition, *Amborella trichopoda*, one of the basal groups of angiosperms, was selected as an outgroup. Based on gene family clustering analysis, 29,189 gene families were identified, of which 7296 were shared by all 12 species, and 485 of these shared families were single-copy gene families (Figure 6). Gene family numbers were compared between *T. wilfordii* and four fabid species. As shown in Figure 7A,

**Table 10.** Summary of short reads coverage of genome assembly.

| | | % of Percentage |
|---|---|---|
| Reads | Mapping rate (%) | 95.31 |
| | Average sequencing depth | 63.05 |
| | Coverage (%) | 93.99 |
| Genome | Coverage at least 4X (%) | 92.32 |
| | Coverage at least 10X (%) | 90.59 |
| | Coverage at least 20X (%) | 87.59 |

**Table 11.** EST evaluation results of the *T. wilfordii* genome.

| Dataset | Number | Total length (bp) | Sequences Covered by assembly (%) | With >90% sequence in one scaffold | | With > 50% sequence in one scaffold | |
|---|---|---|---|---|---|---|---|
| | | | | Number | Percent (%) | Number | Percent (%) |
| >0 bp | 880,651 | 841,600,565 | 88.384 | 722932 | 82.091 | 773641 | 87.849 |
| >200 bp | 880,651 | 841,600,565 | 88.384 | 722932 | 82.091 | 773641 | 87.849 |
| >500 bp | 461,009 | 712,065,253 | 96.962 | 412738 | 89.529 | 444782 | 96.48 |
| >1000 bp | 293,219 | 592,228,395 | 98.869 | 267886 | 91.36 | 288713 | 98.463 |
| >2000 bp | 114,372 | 333,794,368 | 99.71 | 105639 | 92.364 | 113645 | 99.364 |
| >5000 bp | 4,290 | 26,441,647 | 99.627 | 3914 | 91.235 | 4241 | 98.858 |

**Table 12.** Summary of *de novo* genome assembly and annotation of *T. wilfordii*.

| Genome assembly | Number | Metric |
|---|---|---|
| Total contigs | 553 | 340.12 Mb |
| Contig N50 | 34 | 3.09 Mb |
| Total scaffolds | 470 | 342.59 Mb |
| Scaffold N50 | 25 | 5.43 Mb |
| GC content (%) | | 37.19 |
| Heterozygosity rate (%) | | 0.25 |
| Pseudochromosomes | 23 | 311.85 Mb |
| Super-scaffold N50 | | 13.03 Mb |
| **Genome annotation** | | |
| Repetitive sequences | 44.31% | 151.81 Mb |
| Noncoding RNAs | 4770 | 0.82 Mb |
| Structure genes | 31593 | 100.48 Mb |

**Table 13.** Statistics of Hi–C assembly.

| Sample ID | Contig length | Scaffold length | Contig number* | Scaffold number |
|---|---|---|---|---|
| Total | 340,124,188 | 342,608,929 | 566 | 279 |
| Max | 7,962,777 | 17,748,360 | - | - |
| Number ≥ 2000 | - | - | 566 | 279 |
| N50 | 2,929,360 | 13,028,512 | 36 | 12 |
| N60 | 2,234,477 | 12,513,880 | 49 | 15 |
| N70 | 926,087 | 12,436,525 | 73 | 17 |
| N80 | 332,138 | 11,980,432 | 140 | 20 |
| N90 | 202,098 | 10,152,376 | 271 | 23 |

*Contigs > 100 bp were selected for statistics.

10,722 gene families were shared by *G. max*, *C. sativus*, *G. uralensis*, and *M. truncatula*, and 1086 gene families were specific to *T. wilfordii*. Compared with the most recent common ancestor (MRCA) of the 12 plant species, 15 gene families with 152 genes were expanded, including CYPs (see GigaDB for table [87]), and 42 gene families with 54 genes were

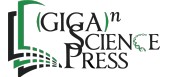

**Table 14.** Scaffold number and length grouped on pseudochromosomes.

| Group | Number of scaffold | Total Length (bp) |
|---|---|---|
| Group1 | 2 | 13,014,687 |
| Group2 | 4 | 11,285,713 |
| Group3 | 9 | 13,837,602 |
| Group4 | 19 | 16,052,000 |
| Group5 | 7 | 15,585,988 |
| Group6 | 12 | 16,997,832 |
| Group7 | 4 | 12,198,226 |
| Group8 | 11 | 15,126,992 |
| Group9 | 8 | 17,748,360 |
| Group10 | 6 | 16,856,595 |
| Group11 | 9 | 10,152,376 |
| Group12 | 8 | 13,417,622 |
| Group13 | 12 | 12,513,880 |
| Group14 | 17 | 14,247,452 |
| Group15 | 12 | 12,206,946 |
| Group16 | 14 | 13,028,512 |
| Group17 | 12 | 13,091,630 |
| Group18 | 5 | 12,436,525 |
| Group19 | 12 | 12,453,453 |
| Group20 | 9 | 12,746,742 |
| Group21 | 8 | 13,078,104 |
| Group22 | 15 | 11,980,432 |
| Group23 | 13 | 11,790,904 |
| Total | 228 | 311,848,573 (91.02%) |

**Table 15.** Summary of gene structure annotation.

| | Gene set | Number | Average transcript length (bp) | Average CDS length (bp) | Average exons per gene | Average exon length (bp) | Average intron length (bp) |
|---|---|---|---|---|---|---|---|
| *De novo* | Augustus | 28,686 | 3,220.73 | 1,220.87 | 5.18 | 235.80 | 478.71 |
| | GlimmerHMM | 48,445 | 5,104.71 | 779.48 | 3.35 | 233.02 | 1,844.39 |
| | SNAP | 38,054 | 2,620.07 | 821.85 | 4.29 | 191.65 | 546.86 |
| | Geneid | 41,284 | 4,097.53 | 941.16 | 4.72 | 199.31 | 848.01 |
| | Genscan | 27,524 | 7,575.88 | 1,391.54 | 6.72 | 207.03 | 1,080.88 |
| Homolog | Rco | 25,778 | 2,920.77 | 1,170.30 | 5.07 | 230.82 | 430.08 |
| | Gur | 24,898 | 2,858.12 | 1,121.45 | 4.87 | 230.21 | 448.59 |
| | Vvi | 25,299 | 2,873.47 | 1,140.76 | 5.03 | 226.77 | 429.90 |
| | Ath | 24,236 | 2,814.33 | 1,131.59 | 4.94 | 229.01 | 426.97 |
| | Csa | 25,267 | 2,748.49 | 1,125.37 | 4.86 | 231.64 | 420.68 |
| | Mtr | 25,346 | 2,771.60 | 1,111.88 | 4.87 | 228.30 | 428.84 |
| RNASeq | PASA | 123,067 | 3,258.88 | 1,066.20 | 5.26 | 202.69 | 514.67 |
| | Transcripts | 48,394 | 6,624.21 | 2,133.18 | 6.71 | 317.76 | 786.09 |
| EVM | | 34,739 | 3,020.48 | 1,106.33 | 4.91 | 225.17 | 489.13 |
| Pasa-update* | | 34,427 | 3,016.92 | 1,130.24 | 4.97 | 227.63 | 475.81 |
| Final set* | | 31,593 | 3,180.62 | 1,182.78 | 5.22 | 226.73 | 473.80 |

* Containing UTR region. Rco: *Ricinus communis*; Gur: *Glycyrrhiza uralensis*; Vvi: *Vitis vinifera*; Ath: *Arabidopsis thaliana*; Csa: *Cucumis sativus*; Mtr: *Medicago truncatula*.

contracted in *T. wilfordii* (Figure 7B). KEGG analysis showed that the expanded genes were enriched in pathways related to 'ubiquinone and other terpenoid-quinone biosynthesis' and 'steroid hormone biosynthesis', suggesting that gene family expansion contributed to specialized metabolite biosynthesis in *T. wilfordii*. A phylogenetic tree was constructed based on the super-alignment matrix of 485 single-copy orthologous genes from the

**Table 16.** Summary of gene function annotation.

|  | Number | Percent(%) |
|---|---|---|
| Total | 31,593 | - |
| Swissprot | 25,392 | 80.40 |
| Nr | 30,388 | 96.20 |
| KEGG | 24,509 | 77.60 |
| InterPro | 29,587 | 93.70 |
| GO | 17,963 | 56.90 |
| Pfam | 24,502 | 77.60 |
| Annotated | 30,535 | 96.70 |
| Unannotated | 1,058 | 3.30 |

**Table 17.** Summary of repetitive sequences.

|  | Denovo+Repbase* | | TE Proteins** | | Combined TEs*** | |
|---|---|---|---|---|---|---|
|  | Length (bp) | % in Genome | Length (bp) | % in Genome | Length (bp) | % in Genome |
| DNA | 5,485,526 | 1.60 | 464,373 | 0.14 | 5,755,303 | 1.68 |
| LINE | 2,270,719 | 0.66 | 161,136 | 0.05 | 2,365,709 | 0.69 |
| SINE | 661,709 | 0.19 | 0 | 0 | 661,709 | 0.19 |
| LTR | 124,801,666 | 36.43 | 24,680,784 | 7.20 | 125,876,630 | 36.74 |
| Unknown | 17,022,433 | 4.97 | 0 | 0 | 17,022,433 | 4.97 |
| Total | 147,540,202 | 43.06 | 25,305,396 | 7.39 | 148,553,910 | 43.36 |

*Denovo+Repbase: Integrated the results of RepeatModeler, RepeatScout, Piler, LTR FINDER and RepBase, then annotated by RepeatMasker; **TE Proteins: annotated by RepeatProteinMask based on RepBase; ***Combined TEs: Integrated the results of Denovo+Repbase and TE Proteins, and then removed redundant.

**Table 18.** Summary of noncoding RNA.

|  | Type | Copy number | Average length (bp) | Total length (bp) | % of genome |
|---|---|---|---|---|---|
|  | miRNA | 355 | 120.26 | 42,694 | 0.012461 |
|  | tRNA | 797 | 75.45 | 60,134 | 0.017552 |
| rRNA | rRNA | 827 | 307.68 | 254,452 | 0.074269 |
|  | 18S | 271 | 690.83 | 187,216 | 0.054644 |
|  | 28S | 293 | 125.56 | 36,790 | 0.010738 |
|  | 5.8S | 104 | 131.08 | 13,632 | 0.003979 |
|  | 5S | 159 | 105.75 | 16,814 | 0.004908 |
| snRNA | snRNA | 982 | 109.09 | 107,128 | 0.031268 |
|  | CD-box | 777 | 101.82 | 79,113 | 0.023091 |
|  | HACA-box | 90 | 129.56 | 11,660 | 0.003403 |
|  | splicing | 114 | 142.35 | 16,228 | 0.004737 |
|  | scaRNA | 1 | 127 | 127 | 0.000037 |

12 species. The branching order showed that *A. thaliana* (Brassicales) and *C. sinensis* (Sapindales) were sister to *P. trichocarpa*, *R. communis* (Malpighiales) and *T. wilfordii* (Celastrales), which diverged approximately 109.1 Mya, followed by divergence of *T. wilfordii* and species from Malpighiales approximately 102.4 Mya (Figures 5B and 8). These results were consistent with a previously proposed phylogenetic order, in which Celastrales and Malpighiales were found to be sister to each other [88].

## Genome-wide identification and analysis of CYP candidates involved in celastrol biosynthesis

The candidate gene identification procedure is illustrated in Figure 9. Based on HMMER analysis, 213 full-length ORFs of *CYP* genes were extracted from the *T. wilfordii* genome;

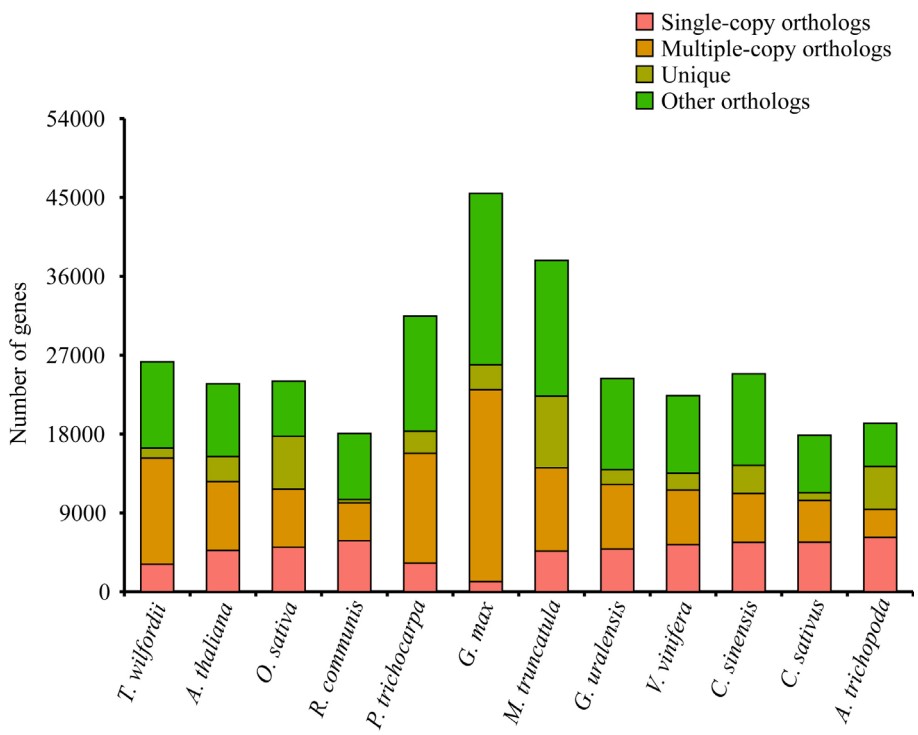

**Figure 6.** The distribution of genes in different species.

these were annotated phylogenetic analysis with *CYPs* from *A. thaliana*. Thirty-five *CYPs* related to triterpenoid oxidases were identified, belonging to different subfamilies, including CYP716, CYP72, CYP71, CYP93, CYP705A and CYP81, which were previously reported to be functionally associated with diverse triterpenoid structural modifications (Figure 10) [89].

On the other hand, the expression patterns of the 213 identified *CYPs* were identified with *TwOSC1* and *TwOSC3*, which are the two committed enzymes involved in the biosynthesis of the precursor of celastrol in *T. wilfordii* [21]. Based on RNA-seq data for various tissues, 20 profiles of gene coexpression were obtained, of which only profiles #3 and #13 showed significance (*P*-value < 0.05) (Figure 11). Profile #3 contained *TwOSC3*, and 45 *CYPs* showed similar expression patterns, while profile #13 included *TwOSC1*, and 51 *CYPs* had coexpression trends (Figure 12). This suggests that these *CYPs* are potentially involved in the biosynthesis of celastrol.

To narrow down the candidate genes, the 35 *CYPs* identified by phylogenetic analysis were compared, and the genes showed patterns of coexpression with *TwOSC1* and *TwOSC3*. As shown in Figure 13A, nine and seven triterpenoid biosynthesis-related *CYPs* showed patterns of coexpression with *TwOSC1* and *TwOSC3*, respectively. However, no *CYPs* were common between the *TwOSC1* group and *TwOSC3* group. Based on tissue expression profiles, the 16 *CYPs* were clustered separately into two clades with *TwOSC1* and *TwOSC3*, in which *TwOSC3* exhibited root-specific expression, while *TwOSC1* was highly expressed in leaves and other aerial parts (Figure 13B).

Gene-to-gene and gene-to-metabolite Pearson's correlation coefficients (r) were calculated using the tissue expression profiles of the 16 outstanding *CYPs* mentioned above,

A

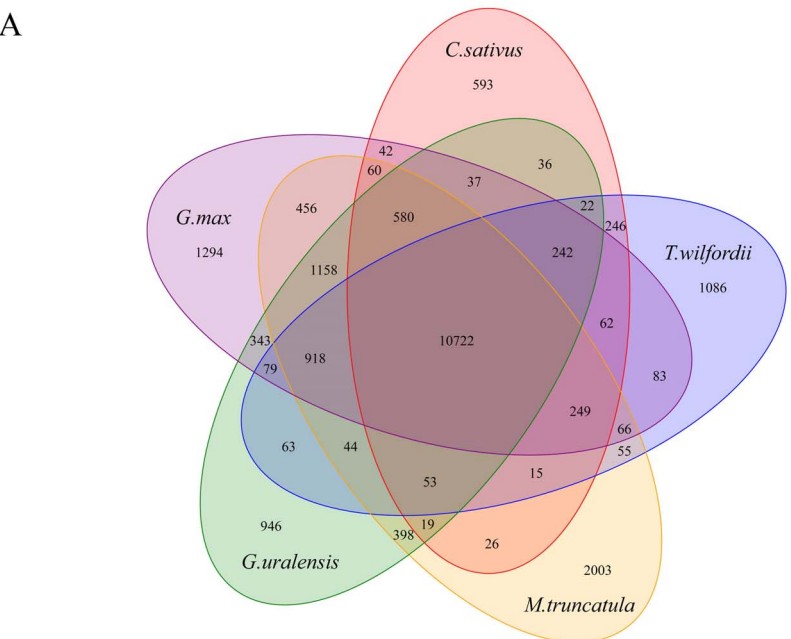

B

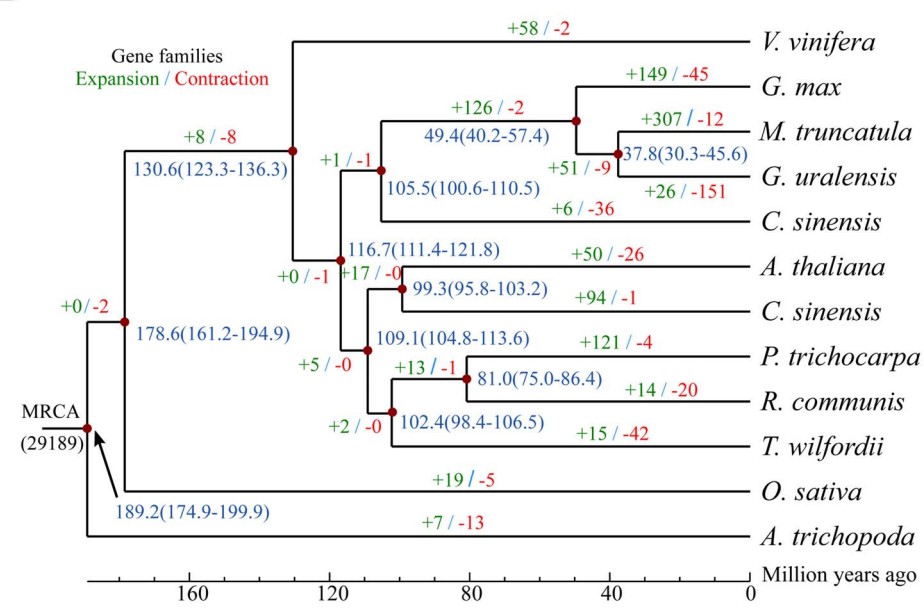

**Figure 7.** Comparative genomic analysis. (A) Venn diagram of common and unique gene families in *Tripterygium wilfordii* with those in other species. (B) Phylogenetic analysis, divergence time estimation, and gene family expansions and contractions. Divergence times (Mya) are indicated by blue numbers, and numbers in brackets represent confidence intervals. Gene family expansions and contractions are indicated by green and red numbers, respectively.

as well as three other known genes related to celastrol biosynthesis (*TwHMGR1*, *TwFPS1* and *TwDXR*) [90–92], and the known intermediate product and celastrol concentrations [21]. As shown in Figure 13C, seven *CYPs* positively correlated with celastrol biosynthesis-related genes, with high Pearson's r and significant *P*-values. In addition, these *CYPs* highly

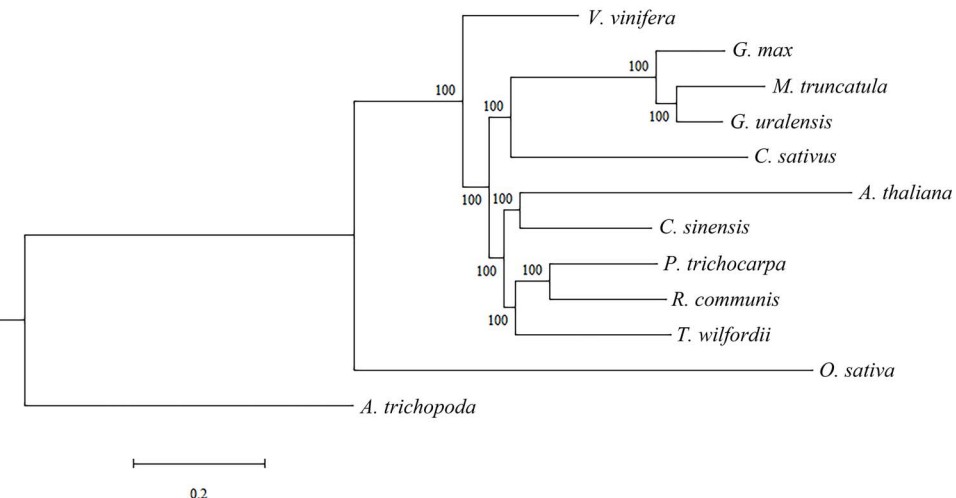

**Figure 8.** Phylogenetic tree of *Tripterygium wilfordii* and other selected species. The branch length represents the evolution rate, and the value on the branch represents the value of bootstrap support.

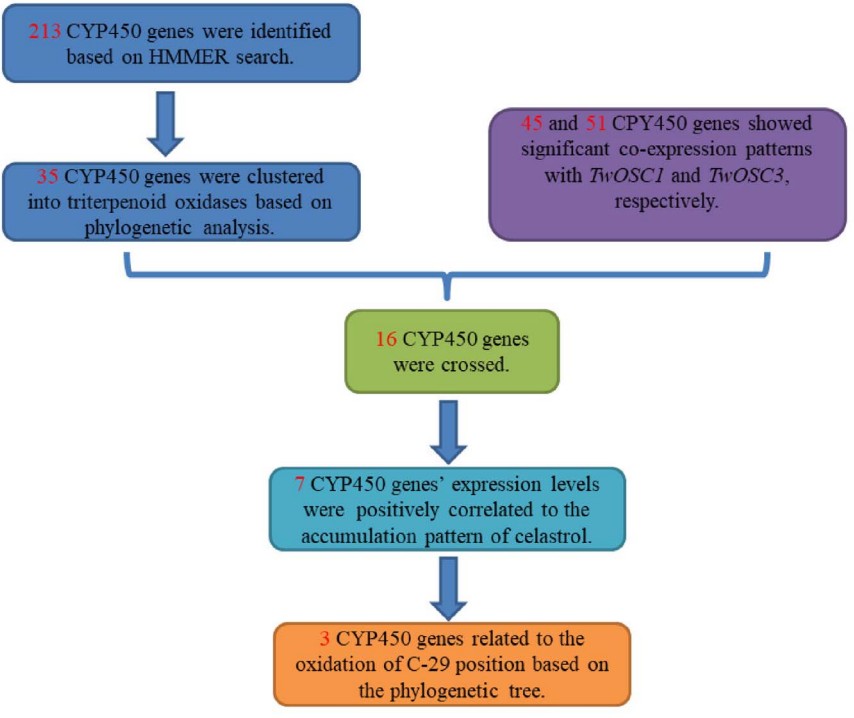

**Figure 9.** The procedure of candidate CYP450 gene identification.

correlated with the levels of 29-hydroxy-friedelan-3-one, polpunonic acid, wilforic acid A and celastrol, which all specifically accumulated in the roots of *T. wilfordii* (Figure 14).

Phylogenetic analysis placed these seven *CYPs* and *CYPs* from other species into three clades representing different functions in the structural modifications of triterpenoids

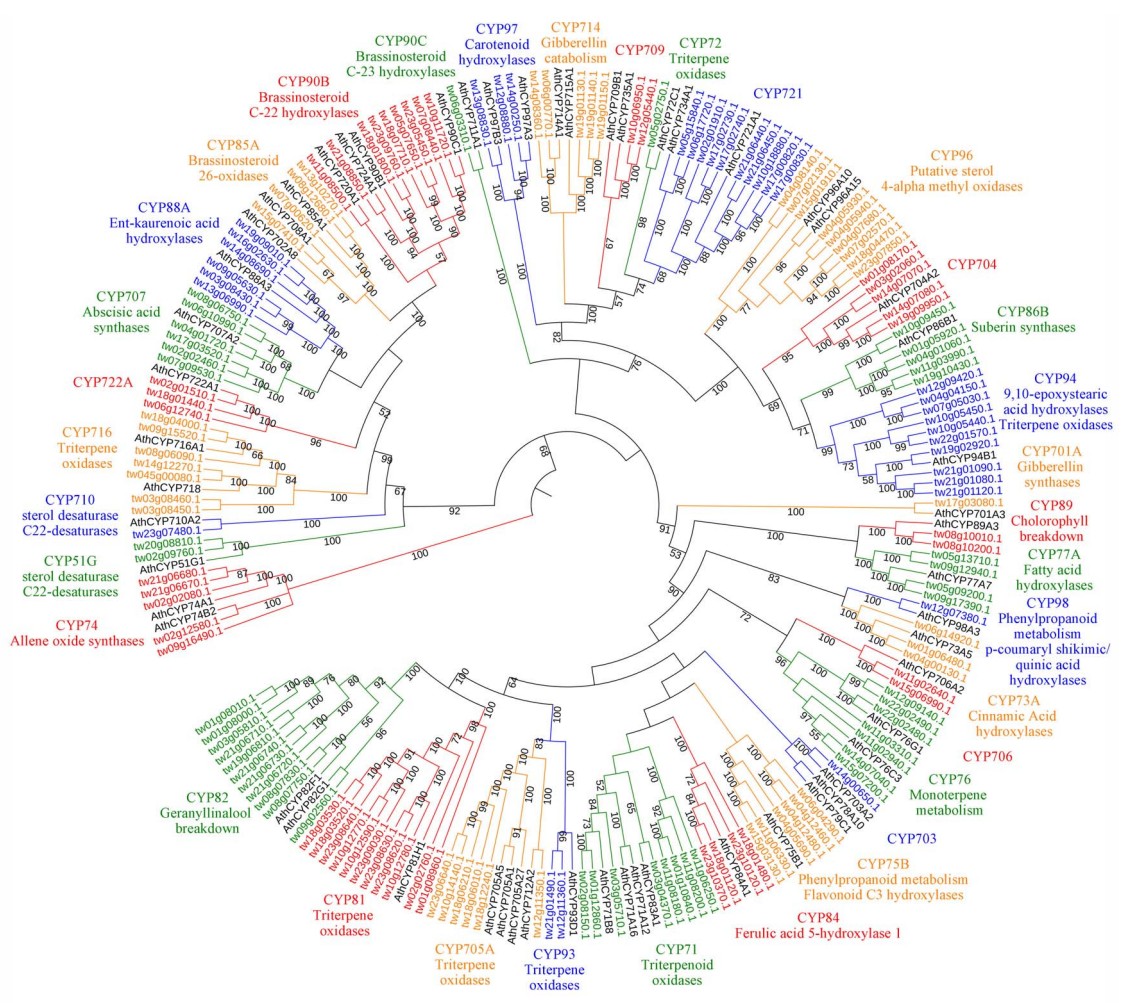

**Figure 10.** Phylogenetic tree of *Tripterygium wilfordii* CYPs. Diverse functions of CYPs are annotated by CYPs from *Arabidopsis thaliana*. A phylogenetic tree was built using the neighbor-joining method with a bootstrap test (*n* = 1000 replications). Numbers on the branches represent bootstrap support values.

(Figure 13D). Two CYPs (tw18g03520.1 and tw01g08960.1) were clustered with CYP81Q58 from *C. sativus*, which catalyzes hydroxylation of the C-25 position in cucurbitacins [93]; four CYPs (tw12g11350.1, tw18g06010.1, tw18g06210.1 and tw23g06640.1) were clustered with CYP712K4 from *Monteverdia ilicifolia*, which catalyzes the oxidation of the C-29 position using friedelin as a substrate [94]; and tw03g08450.1 was clustered with CYP716C11 from *Centella asiatica*, which hydroxylates the C-2 position of oleanolic acid and ursolic acid [95].

Since we were interested in identifying a C-29 position oxidase that could catalyze the conversion of friedelin to polpunonic acid, 3 *CYPs* were finally chosen as candidates for functional validation: tw18g06010.1, tw18g06210.1 and tw23g06640.1 (hereafter referred to as *TwCYP712K1*, *TwCYP712K2* and *TwCYP712K3*) according to the closer relationship with CYP712K4 from *M. ilicifolia*, which is related to C-29 hydroxylation [94].



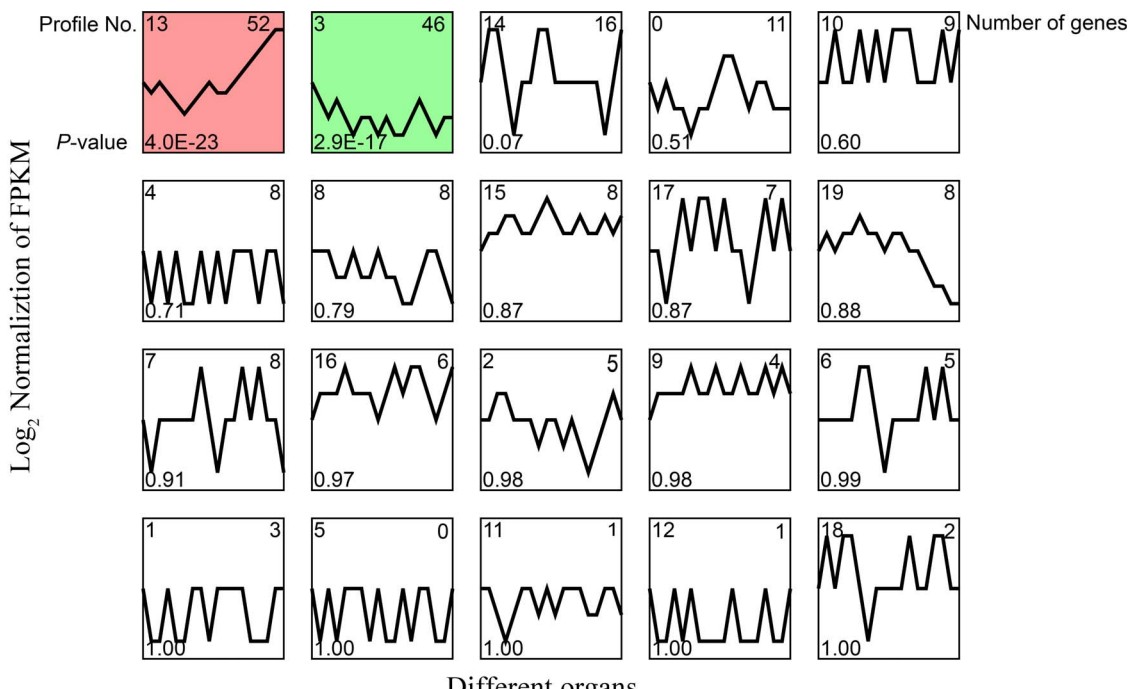

**Figure 11.** Maps of expression trends of CYPs with TwOSC1 and TwOSC3. Profiles ordered based on the *P*-value significance of number of genes assigned versus expected. Numbers on the top left corner represent the profiles number; numbers on the left bottom represent the *P*-value; numbers on the top right corner represent the total number of genes. Colored maps represent the significant enrichment with *P*-value < 0.05.

## Heterologous expression and characterization of putative CYPs

The full-length ORFs of *TwCYP712K1*, *TwCYP712K2* and *TwCYP712K3* were successfully clones and separately expressed in yeast fed with friedelin or 29-hydroxy-friedelan-3-one. However, no new peaks could be detected from the yeast strains expressing the enzymes and supplemented with friedelin compared with the empty vector (EV) control (see GigaDB [87]). This probably because the hydrophobic substrate could not be transported into the yeast cells. When fed with 29-hydroxy-friedelan-3-one, both TwCYP712K1 and TwCYP712K2 converted the substrate to a new compound possessing the same mass charge ratio (m/z) as the polpunonic acid standard, while TwCYP712K3 showed no such activity in this assay (Figure 15A and C). To further explore the enzyme activities, microsomes were extracted from yeast cells and the proteins incubated with friedelin or 29-hydroxy-friedelan-3-one for 12 h. As shown in Figure 15B, TwCYP712K1 and TwCYP712K2 converted 29-hydroxy-friedelan-3-one to a new compound *in vitro*, consistent with previous yeast *in vivo* assays. In addition, no new peak was detected from the enzyme reactions supplemented with friedelin compared with the EV control (see GigaDB [87]).

## Evolutionary analyses of *TwCYP712K1* and *TwCYP712K2*

Genome analysis showed that *TwCYP712K1* (tw18g06010.1) and *TwCYP712K2* (tw18g06210.1) were located in pseudochromosome 18 within an approximately 200-kb region. This led us to examine the evolutionary relationship between specific CYPs. We hypothesized that a gene duplication event must have occurred during evolution. However, amino acid alignment indicated only 70.57% identity between TwCYP712K1 and



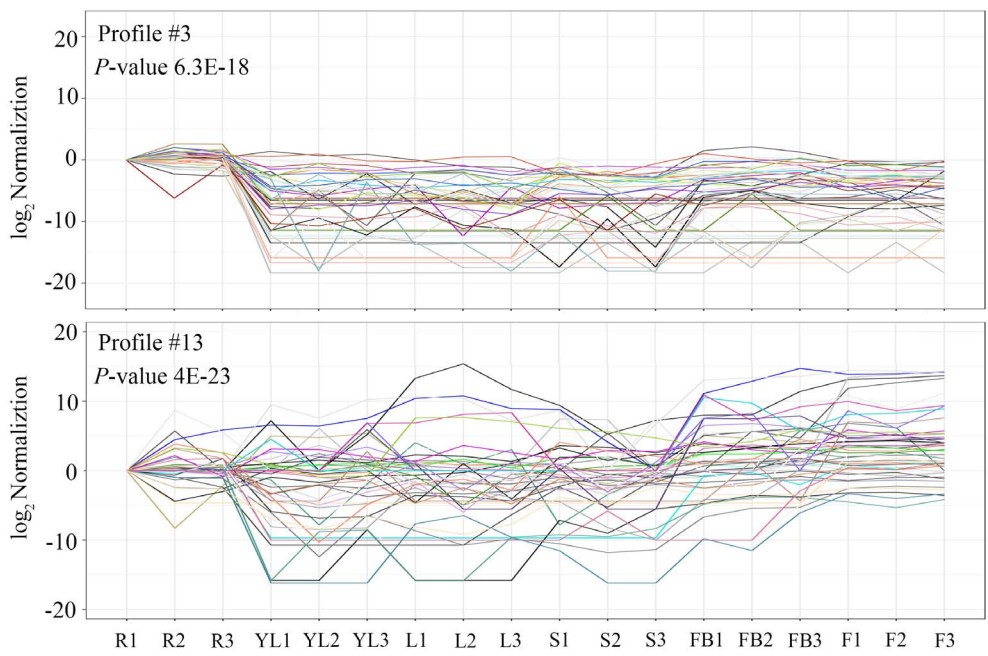

**Figure 12.** Co-expression of potential CYP genes with TwOSC1 and TwOSC3. The upper map indicates the similar expression patterns of *CYPs* with *TwOSC3* and the lower map indicates the similar expression patterns of *CYPs* with *TwOSC1*. R, root; YL, young leaf; L, leaf; S, stem; FB, flower bud; F, flower; numbers 1–3 represent three biological replicates, respectively.

TwCYP712K2 (Figure 16), suggesting that these two genes became specialized a long time ago. Syntenic analysis showed that *TwCYP712K1* and *TwCYP712K2* had corresponding collinear genes in *V. vinifera* but not in *O. sativa*, indicating that these two genes appeared after the species differentiation of Poaceae and Vitales (<178.6 Mya), and they came from the common ancestor but divided after the species differentiation of Vitales (<130.6 Mya) (Figures 7 and 17).

## DISCUSSION

In this study, we provided a high-quality reference genome of *T. wilfordii* with a 340.12 Mb genome assembly (90.5% of the 375.84 Mb estimated genome size) and 3.09 Mb contig N50, and successfully anchored 91.02% of the sequences into 23 pseudochromosomes (Table 12). The quality of our genome is close to that of the recently published *T. wilfordii* genome (348.38 Mb total contigs and 4.36 Mb contig N50), which was sequenced and assembled using Illumina, PacBio and Hi–C sequencing [86]. They also identified a key *CYP* gene that can catalyze the oxidation of a methyl group to the acid moiety of dehydroabietic acid in triptolide biosynthesis, another clinically used specialized metabolite in *T. wilfordii.*

Based on this genomic data, 35 *CYP* genes related to triterpenoid structure modification were identified according to phylogenetic analysis (Figure 8); 16 of these were co-expressed with *TwOSC1* or *TwOSC3* according to tissue-specific transcript profiles (Figures 11 and 12). These genes could be divided into two groups: the *TwOSC1* group was highly expressed in leaves or other aerial parts, and the *TwOSC3* group was specifically expressed in roots (Figure 13B), suggesting a sub-functionalization of *TwOSC1* and *TwOSC3* at the expression level in mediating the biosynthesis of friedelane-type triterpenoids. Correlation coefficient



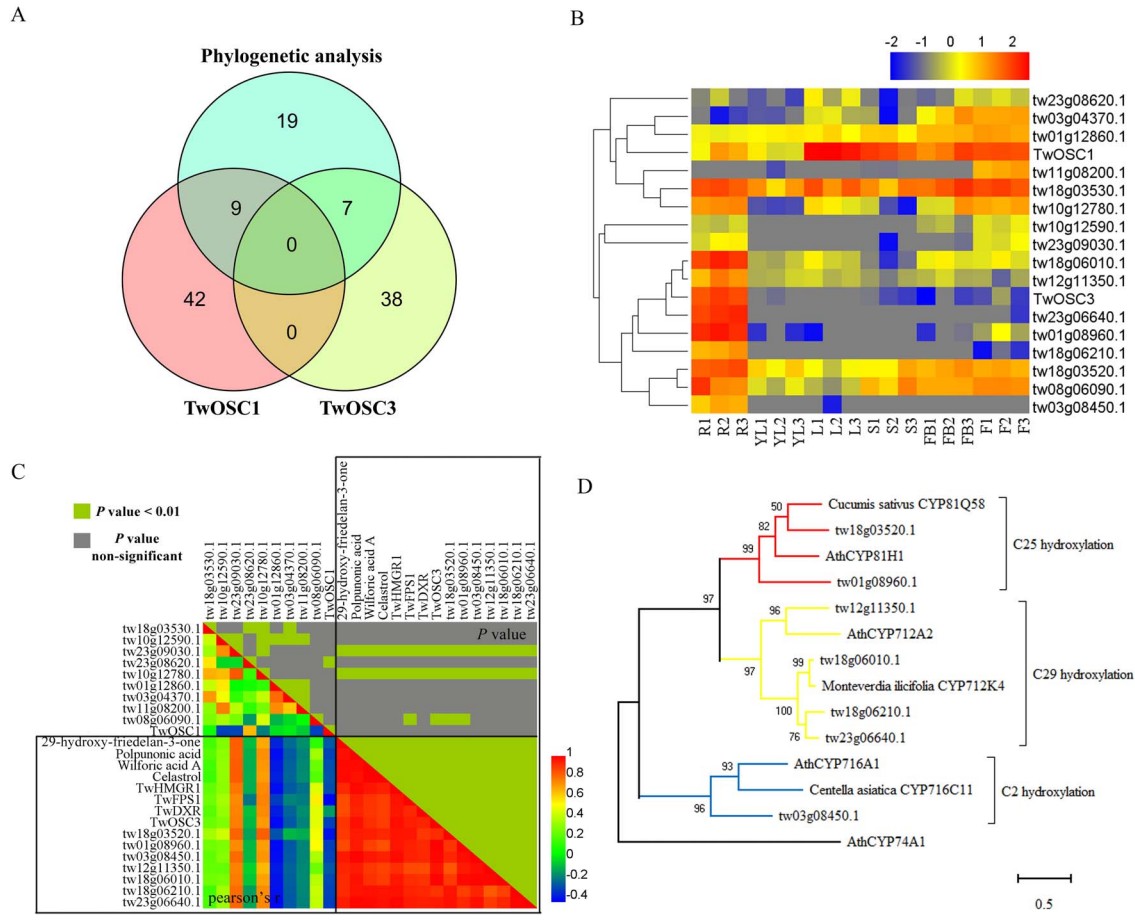

**Figure 13.** Identification of polpunonic acid producing *CYPs* via integration analysis. (A) Venn diagram of CYPs identified by phylogenetic analysis versus co-expression patterns. (B) Tissue-specific expression profiles and clustering of *CYPs* with *TwOSC1* and *TwOSC3*. The gradient bar represents the expression levels from high (red) to low (blue) with log2 normalization, and the gray color represents the empty value. R, root; YL, young leaf; L, leaf; S, stem; FB, flower bud; F, flower. The numbers 1–3 represent three biological replicates. (C) Matrix of Pearson's correlation coefficient and corresponding *P*-value of compounds, biosynthesis-related genes and *CYP* candidates. The lower triangle matrix represents Pearson's correlation coefficient, and the upper triangle matrix presents *P*-values (green indicates *P*-values < 0.01, and gray indicates nonsignificant correlation). The gradient bar represents the correlation coefficient of positive or negative correlation from high to low. The black boxes enclose the highly correlated genes or compounds. (D) Phylogenetic analysis of putative CYPs in celastrol biosynthesis and CYPs known to catalyze structural modifications on triterpenoid scaffolds A phylogenetic tree was built using the maximum-likelihood method with a bootstrap test (*n* = 1000 replications). Allene oxide synthases AtCYP74A1 was set as an outgroup.

testing revealed that the expression levels of six *CYPs* significantly correlated with the expression patterns of genes involved in celastrol biosynthesis and the accumulation patterns of celastrol and its biosynthetic intermediates (Figure 13C). A more subdivided phylogenetic tree showed that three putative *CYPs* were clustered close to *CYP712K4*, which was cloned from *M. ilicifolia* (Figure 13D), another plant belonging to the Celastraceae family. *CYP712K4* encodes an enzyme that catalyzes the oxidation of the C-29 position of friedelin to produce polpunonic acid [94].

Both *in vivo* and *in vitro* assays revealed that TwCYP712K1 and TwCYP712K2 could use 29-hydroxyfriedelan-3-one as a substrate to produce a new compound as the only product, indicating the oxidation of 29-hydroxyfriedelan-3-one at the C-29 position catalyzed by CYPs (Figure 15D). However, the peak of product from reactions of TwCYP712K1 and TwCYP712K3 were deviated with the peak of polpunonic acid standard. To demonstrate

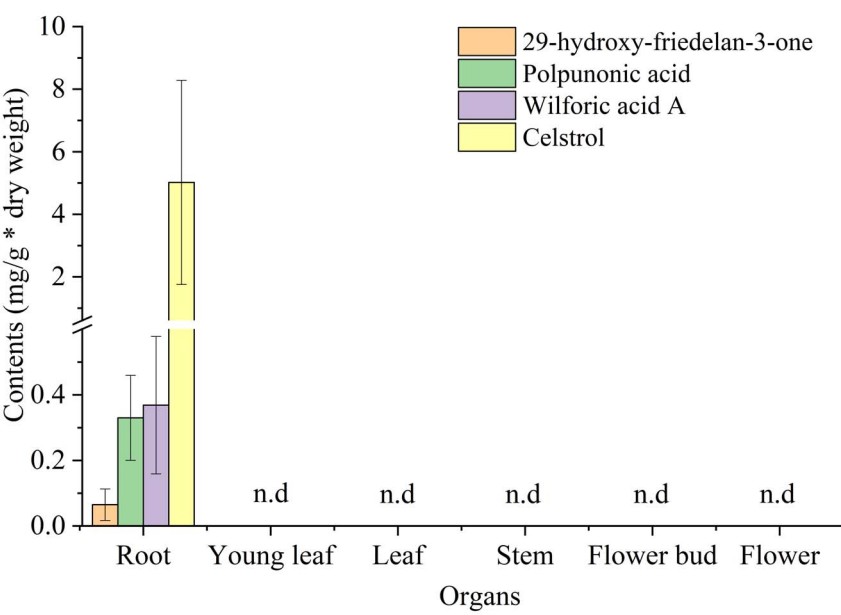

**Figure 14.** The accumulation of celastrol and intermediate products in different tissues of *T. wilfordii.* Bars are means ± SD from three independent biological replicates, n.d=not detected in our experimental condition; the value of not detected compounds was set to 0.

that these were same compound, future experiments will need to add another reaction containing 29-hydroxyfriedelan-3-one, buffer, enzyme and polpunonic acid standard (small amount, mixed into the reaction before LC-MS) in our follow on research. Comparative genome analysis showed that *TwCYP712K1* and *TwCYP712K2* derived from a common ancestor (Figure 17). Although they catalyzed the same reaction and were located close to each other on the same chromosome, the identity of the amino acid sequence (70.57%) was not high (Figure 16). This suggests that *TwCYP712K1* and *TwCYP712K2* did not come from recent gene duplication, but separated during the evolution of the Celastraceae family. Interestingly, important catalytic activity for polpunonic acid biosynthesis in *T. wilfordii* was conserved in both these enzymes. As more genomes of the Celastraceae family are released, further evolutionary details of *TwCYP712K1* and *TwCYP712K2* can be investigated. There are many reports of genes encoding certain natural product pathways being grouped together in gene clusters to catalyze the biosynthesis of plant specialized metabolism, including triterpenoids [93, 96, 97]. However, neither the CYPs we identified, nor the signature enzymes TwOSC1 (tw21g04301.1) and TwOSC3 (tw20g03871.1), were clustered together.

## POTENTIAL FOR REUSE

We reported the reference genome assembly of *T. wilfordii* and provided a useful strategy for screening the genes involved in plant specialized metabolism. For further exploration, the genome can be used for comparative genomic analyses; for example, to resolve the controversial phylogenetic relationships within the COM (Celastrales, Oxalidales and Malpighiales) clade [88]. Additionally, full-length transcriptome and tissue-specific RNA-seq data can be used to mine all the biosynthetic pathway genes of celastrol, as well as the biosynthetic pathways of the diterpenoid and alkaloid active ingredients.

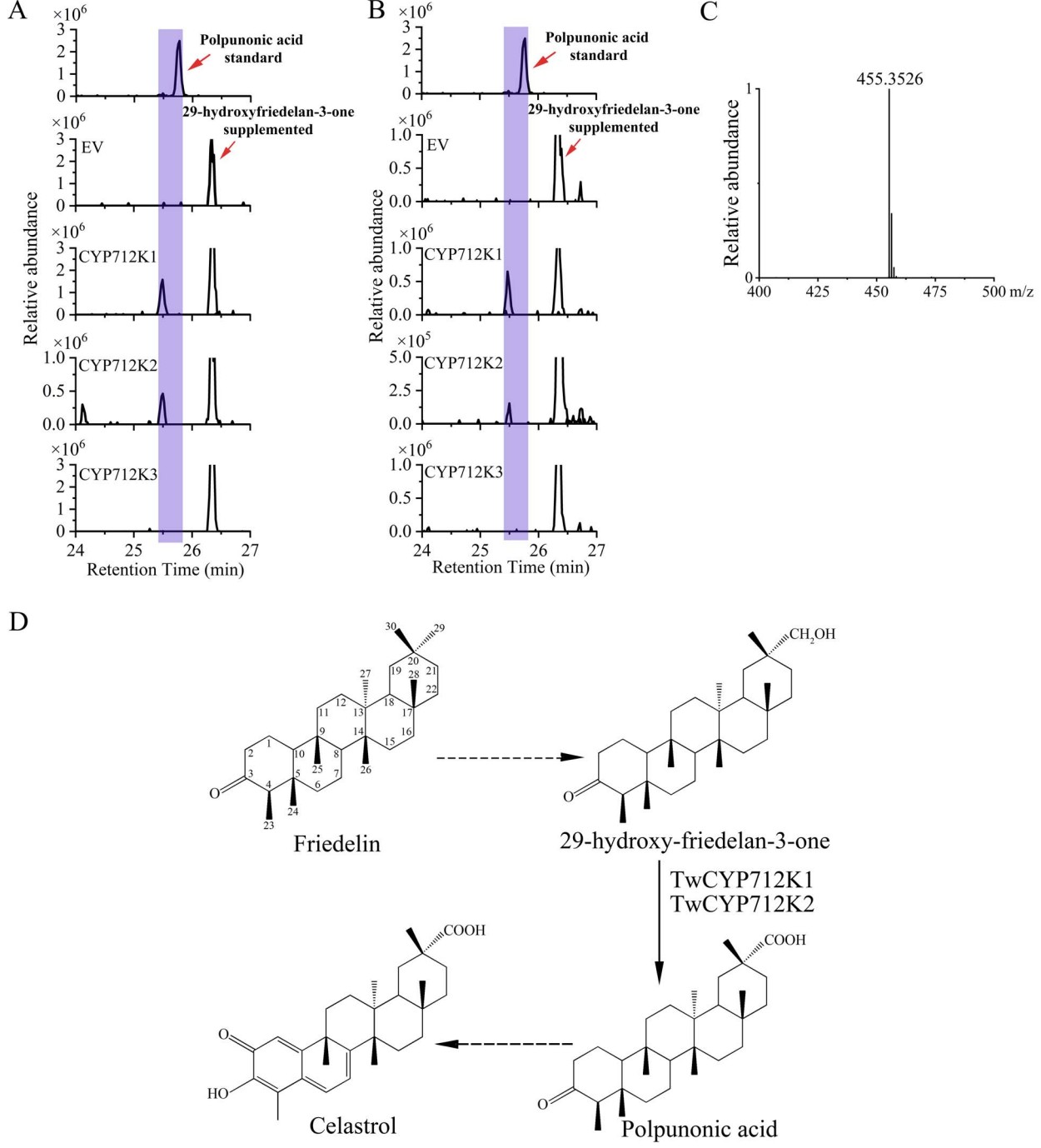

**Figure 15.** Characterization of candidate CYPs for polpunonic acid biosynthesis. (A) Liquid chromatography–mass spectrometry (LC–MS) analyses of yeast samples fed with 29-hydroxy-friedelan-3-one as a substrate *in vivo*. Top, polpunonic acid standard; EV, empty vector; TwCYP712K1-TwCYP712K3, yeast expressing the corresponding proteins. New peaks with the same retention time as polpunonic acid are highlighted. (B) LC–MS analyses of microsome samples incubated with 29-hydroxy-friedelan-3-one as a substrate *in vitro*. Top, polpunonic acid standard; EV, empty vector; TwCYP712K1-TwCYP712K3, corresponding microsomes extracted from yeast cells. New peaks with the same retention time as polpunonic acid are highlighted. (C) Accurate masses of polpunonic acid. (D) Putative oxidation of 29-hydroxy-friedelan-3-one catalyzed by TwCYP712K1 and TwCYP712K2. The dashed arrow indicates multiple catalyzed steps that were unidentified.

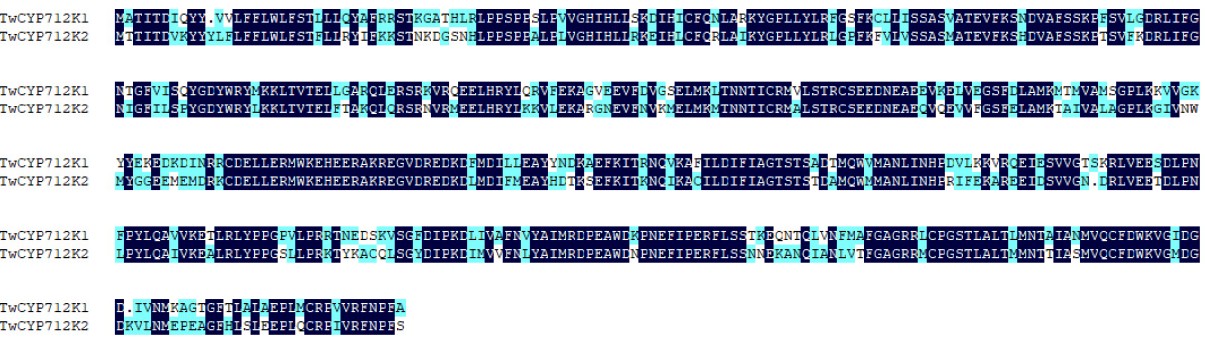

**Figure 16.** Alignment of TwCYP712K1 and TwCYP712K2 sequences. TwCYP712K1 and TwCYP712K2 exhibited 70.57% identity. The consensus sequences were highlighted by deep blue color.

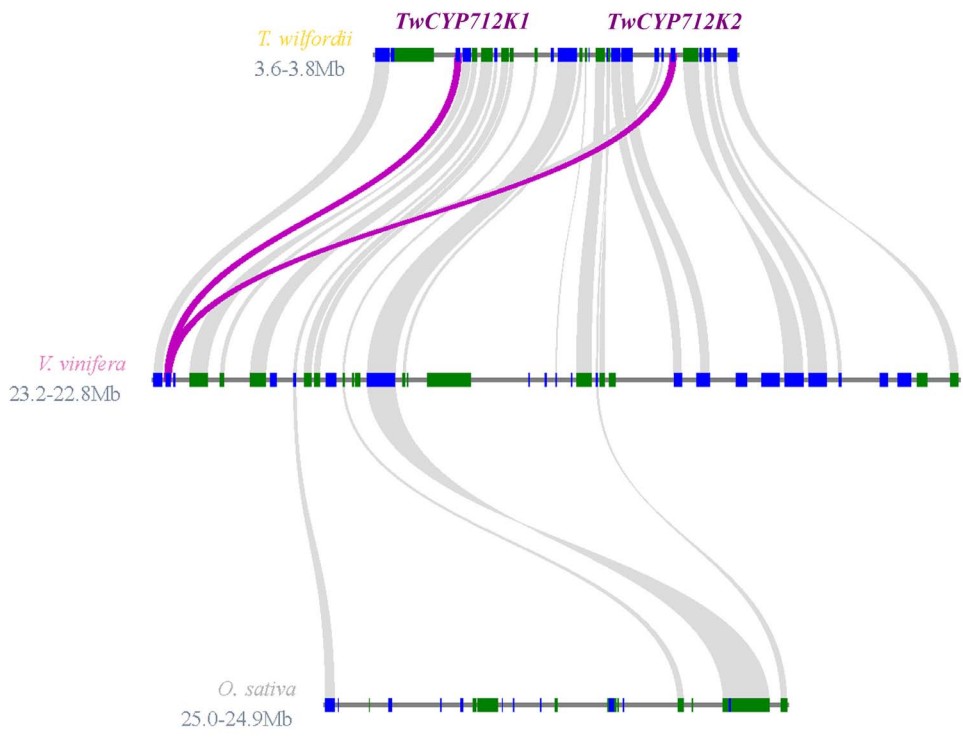

**Figure 17.** Syntenic analysis of *TwCYP712K1* and *TwCYP712K2* genes. Focused *CYP* genes are colored purple.

## DECLARATIONS
### ETHICS APPROVAL AND CONSENT TO PARTICIPATE
Not applicable.

### CONSENT FOR PUBLICATION
Not applicable.

## AVAILABILITY OF DATA AND MATERIALS

The datasets generated and analyzed during the current study are available in GenBank of the National Center for Biotechnology Information (NCBI), under the BioProject number PRJNA640746. Gene and protein sequences of TwCYP712K1 (MT633088) and TwCYP712K2 (MT633089) are deposited in GenBank. Raw mass spectrometry data are available in MetaboLights under the study number MTBLS1080. Additional datasets are available in the *GigaScience* GigaDB repository [87].

## COMPETING INTERESTS

The authors declare that they have no competing interests.

## FUNDING

This work was supported by the National Key R&D Program of China (grant numbers 2018YFC1706202, 2019YFD1000703, 2018YFD1000701, and 2020YFA0907901), the National Natural Science Foundation of China (grant numbers 31870282, 31700268), Youth Innovation Promotion Association of the Chinese Academy of Sciences, and the Chenshan Special Fund for Shanghai Landscaping Administration Bureau Program (grant numbers G182401, G182402, G192419, G192413, G192414 and G202402). Q.Z. is also support by the Shanghai Youth Talent Support Program and SA-SIBS Scholarship Program.

## AUTHORS' CONTRIBUTIONS

T.L.P. and Q.Z initiated the program, coordinated the project, and wrote the manuscript. B.J.G prepared the plant materials. Y.K., J.L., M.Y.C., Y.M.F., and H.F prepared and analyzed the samples. T.L.P., M.X.Y., J.Y. and Q.Z. analyzed the genome sequence. M.X.Y. and J.Y. revised the manuscript. All authors read and approved the final manuscript.

## ACKNOWLEDGEMENTS

We greatly appreciate the experimental facilities and services provided by the office of Chenshan Plant Science Research Center.

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
