## [Reviewer Report]

Comments on revised manuscriptThe authors greatly improved the readability of the manuscript. I would suggest they comment more on how their assembly differs from the existing assembly to clarify the novelty of this work versus the previously published genome.

---

## [Reviewer Report]

Comments on revised manuscriptThe authors have been responded to my comments in the revision of DRR-20200905. The new version of the manuscript has been improved. I would be very happy to recommend it for publication on GigaByte if some extra information were considered, especially the comment 4.  1. In the logically and methodology, HiC anchoring and superscaffold N50 estimating were before structural gene predication, repeat calculation and other comparative evolutionary analyses. Please revise them on line 29-34.  2. It is very advisable that the version of software was supplemented in the revised manuscript, but concomitant parameters or URL of software should be in an additional file rather than in the main manuscript. Overlong parameters or URL will affect reading coherently. I notice that the related comment was suggested by another reviewer, so please comprehensive consideration by authors and editors.   3. Two candidate CYP genes was obtained from co-expression and WGNCA analysis, so Line 376-410 Metabolite analysis might be list before Line 337 Enzyme assays of yeast in vivo.   4. In figure 5, the peak of product from reactions of CYP712K1 and CYP712K3 were deviated with the peak of polpunonic acid standard, error range might be -0.3 min. For proving they were same compound, I strongly recommend adding another reaction [reaction (substracte +buffer +enzyme) +product standard (small amount, mixed into the reaction before LC-MS)] in your future research-related. If the peak of this reaction were same with the reactions of CYP712K1 and CYP712K3, the function of enzyme would be confirmed. On the other side, similar with Figure 5C, the primary and secondary mass spectroscopy of the product peaks from reactions of CYP712K1 and CYP712K3 could be also provide the convincing evidence of products are the same as the standard. So, please provide the MS and MS2 of the reactions of CYP712K1 and CYP712K3 in Figure 5A and Figure 5B as the additional files.   5. Figure 5C was not mentioned in the manuscript.

---

## [Reviewer Report]

Reviewer name and names of any other individual's who aided in reviewer C Robin BuellDo you understand and agree to our policy of having open and named reviews, and having your review included with the published papers. (If no, please inform the editor that you cannot review this manuscript.)YesIs the language of sufficient quality?NoPlease add additional comments on language quality to clarify if needed
The manuscript could be improved with a round of editing for grammar. Are all data available and do they match the descriptions in the paper? YesAdditional CommentsAre the data and metadata consistent with relevant minimum information or reporting standards? See GigaDB checklists for examples <a href="http://gigadb.org/site/guide" target="_blank">http://gigadb.org/site/guide</a>YesAdditional CommentsIs the data acquisition clear, complete and methodologically sound?YesAdditional CommentsIs there sufficient detail in the methods and data-processing steps to allow reproduction?NoAdditional CommentsSee my comments below. The sequencing, assembly and annotation methods need more details. Is there sufficient data validation and statistical analyses of data quality? YesAdditional CommentsIs the validation suitable for this type of data?YesAdditional CommentsIs there sufficient information for others to reuse this dataset or integrate it with other data?YesAdditional CommentsAny Additional Overall Comments to the AuthorThis manuscript describes the sequencing, assembly, annotation, and analysis of the Tripterygium wilfordii genome. T. wilfordii is a medicinal plant that has long been used in traditional medicine due to its production of alkaloids and triterpenoids; the focus of this study was identify cytochrome P450s involved in biosynthesis of the triterpenoid celastrol.  Based on the genome assembly metrics, the authors generated a robust representation of the genome sequence. Improvements in the analyses of the genome and in the manuscript would greatly strengthen confidence in the assembly. The authors should provide these metrics and additional information to the manuscript:  More details on the error correction of the assembly. Based on the methods, both nanopore and Illumina WGS reads were used, however, this is not explicit nor are any metrics of the error correction provided.  Specifically it is not discussed how the nanopore reads were assembled. A company is cited for the genome assembly. Information on what assembly software that was used must be provided.  Every software program used, its version, and the parameters used should be provided in the methods. This is often missing.  The quality of the genome should be confirmed using both alignment of the whole genome shotgun reads and the mRNAseq data. Specific metrics should be provided include: total and percentage of reads that mapped, read pairs that mapped in the correct orientation.  No details on read quality assessment or trimming are provided  The CEGMA results should be omitted, this program has been deprecated.  Line 337: The DNA was sheared not interrupted into fragments Line 343: More details on the library preparation and sequencing for the nanopore reads.   Do the authors know the genome size of the species based on flow cytometry? Do you know the number of chromosomes that this species has? This should be stated and discussed in context of the assembly size and number of pseudochromosomes  The genome wide identification of the CYP450 candidates was difficult to follow. This section should be revised so that it is clear how the authors identified their candidate genes. Potentially adding a supplemental figure would be helpful. I found the coexpression pattern extremely difficult to follow. I would not call coexpression patterns coexpression profiles. Specifically I did not understand the sentence on line 202 “However, no….”. Essentially this is just sub-functionalization at the expression level, not that there are two independent pathways.   The evolution section should be expanded. How divergent are T. wilfordii from P. trichocarpa and R. communis?   Table 1: Index should be replaced with metric  Figure S1: What k-Mer was used in the analysis? Figure S5: Unclear what is on the X or y axis. Expand the figure legend.  The manuscript should be proofed for grammar as there are numerous sentences that need editing. 
RecommendationMajor Revision

---

## [Reviewer Report]

Upload additional filesDRR-20200905/form/Comments.docxReviewer name and names of any other individual's who aided in reviewer Xupo DingDo you understand and agree to our policy of having open and named reviews, and having your review included with the published papers. (If no, please inform the editor that you cannot review this manuscript.)YesIs the language of sufficient quality?NoPlease add additional comments on language quality to clarify if needed
The language of one third paragraph is sufficient qualityAre all data available and do they match the descriptions in the paper? YesAdditional CommentsAre the data and metadata consistent with relevant minimum information or reporting standards? See GigaDB checklists for examples <a href="http://gigadb.org/site/guide" target="_blank">http://gigadb.org/site/guide</a>YesAdditional CommentsIs the data acquisition clear, complete and methodologically sound?YesAdditional CommentsIs there sufficient detail in the methods and data-processing steps to allow reproduction?YesAdditional CommentsIs there sufficient data validation and statistical analyses of data quality? YesAdditional CommentsIs the validation suitable for this type of data?YesAdditional CommentsIs there sufficient information for others to reuse this dataset or integrate it with other data?YesAdditional CommentsAny Additional Overall Comments to the AuthorRecommendationMajor Revision